# MYC reshapes CTCF-mediated chromatin architecture in prostate cancer

Zhao Wei [1,13], Song Wang [2,3,4,13], Yaning Xu [2,3,4], Wenzheng Wang [2,3,4], Fraser Soares [5], Musaddeque Ahmed [5], Ping Su [6], Tingting Wang [7], Elias Orouji [8,9], Xin Xu [5], Yong Zeng [5], Sujun Chen [5], Xiaoyu Liu [2,3,4], Tianwei Jia [2,3,4], Zhaojian Liu [10], Lutao Du [2,3,4], Yunshan Wang [2,3,4], Shaoyong Chen [11], Chuanxin Wang [2,3,4], Housheng Hansen He [5,12,14] ✉ & Haiyang Guo [2,3,4,14] ✉

MYC is a well characterized oncogenic transcription factor in prostate cancer, and CTCF is the main architectural protein of three-dimensional genome organization. However, the functional link between the two master regulators has not been reported. In this study, we find that MYC rewires prostate cancer chromatin architecture by interacting with CTCF protein. Through combining the H3K27ac, AR and CTCF HiChIP profiles with CRISPR deletion of a CTCF site upstream of MYC gene, we show that MYC activation leads to profound changes of CTCF-mediated chromatin looping. Mechanistically, MYC colocalizes with CTCF at a subset of genomic sites, and enhances CTCF occupancy at these loci. Consequently, the CTCF-mediated chromatin looping is potentiated by MYC activation, resulting in the disruption of enhancer-promoter looping at neuroendocrine lineage plasticity genes. Collectively, our findings define the function of MYC as a CTCF co-factor in three-dimensional genome organization.

The oncoprotein *MYC* regulates various biological activities that contribute to tumorigenesis. Numerous studies have described the regulation of *MYC* expression by distal enhancer dynamics[1–3], but the function of *MYC* in three-dimensional (3D) genome organization is largely unexplored. As a master transcription factor (TF), MYC protein has an intrinsically disordered transactivation domain and can form phase-separated condensates with the MED1 protein[4]. *MYC* is also required for the chromatin decompaction and loop formation during B cell activation[5]. A recent study suggests *MYC* overexpression in U2OS

cells increases global chromatin interactions at super-enhancers and MYC binding sites[6]. Although these clues imply *MYC* functionalities in 3D genome organization, the molecular mechanisms have not been specified yet.

*CTCF* is a principal 3D chromatin architecture organizer, which functions together with the cohesin complex to establish chromatin loops and structure topologically associated domains (TAD)[7]. *CTCF* shows pleiotropic functions in gene expression regulation, such as insulating enhancers/promoters within CTCF-CTCF loops from

[1]Department of Clinical Laboratory, Qilu Hospital of Shandong University, Jinan 250012 Shandong Province, China. [2]Department of Clinical Laboratory, the Second Hospital, Cheeloo College of Medicine, Shandong University, Jinan 250033 Shandong, China. [3]Shandong Engineering & Technology Research Center for Tumor Marker Detection, Jinan 250033 Shandong, China. [4]Shandong Provincial Clinical Medicine Research Center for Clinical Laboratory, Jinan 250033 Shandong, China. [5]Princess Margaret Cancer Center/University Health Network, Toronto, Ontario M5G 1L7, Canada. [6]National Administration of Health Data, Jinan 250000, China. [7]Institute of Medical Sciences, the Second Hospital, Cheeloo College of Medicine, Shandong University, Jinan 250033 Shandong, China. [8]Epigenetics Initiative, Princess Margaret Genomics Centre, Toronto, ON, Canada. [9]Department of Genomic Medicine, University of Texas MD Anderson Cancer Center, Houston, TX, USA. [10]Key Laboratory of Experimental Teratology, Ministry of Education, Department of Cell Biology, Cheeloo College of Medicine, Shandong University, Jinan, Shandong 250012, China. [11]Hematology-Oncology Division, Department of Medicine, Beth Israel Deaconess Medical Center and Harvard Medical School, Boston, MA 02215, USA. [12]Department of Medical Biophysics, University of Toronto, Toronto, Ontario M5G 2M9, Canada. [13]These authors contributed equally: Zhao Wei, Song Wang. [14]These authors jointly supervised this work: Hansen He, Haiyang Guo. ✉e-mail: Hansen.He@uhnresearch.ca; haiyang.guo@email.sdu.edu.cn

outside regulatory elements (RE) or facilitating enhancer-promoter interactions by colocalizing with other TFs[7,8]. Despite the importance of *CTCF* in 3D genome, only a few proteins, including cohesin[9,10] and MAZ[11,12], have been identified to stabilize the CTCF-mediated chromatin contact. Considering the widespread function of CTCF sites in transcription regulation, functional interactions between *CTCF* and other master regulators could be potentially exploited by cancer cells as a strategy to fine-tune oncogenic gene expression.

Here, we show that MYC reshapes the chromatin architecture of prostate cancer (PCa) cells through interacting with CTCF protein. By defining H3K27ac, AR and CTCF HiChIP profiles and CRISPR deletion of a CTCF site upstream of MYC, we reveal MYC activation results in genome-wide changes of CTCF looping. Mechanistically, MYC interacts with CTCF protein and increases CTCF chromatin binding affinities at MYC/CTCF common sites. Utilizing multi-omics approaches in an ectopic *MYC* expression model, we reveal that *MYC* represses a subset of target genes involved in neuroendocrine lineage plasticity by potentiating CTCF-mediated chromatin looping. Taken together, this study unravels the role of *MYC* in 3D genome architecture, promoting CTCF-mediated chromatin interactions to regulate PCa gene expression.

## Results

### 3D interaction mapping of regulatory elements in PCa

To systematically characterize RE-associated chromatin interactions in PCa, we performed H3K27ac HiChIP in PCa cell lines 22Rv1 and V16A under full FBS medium culture. To investigate androgen-induced chromatin dynamics, we also performed H3K27ac, AR and CTCF HiChIP in VCaP cells under charcoal-stripped FBS medium with and without androgen (DHT) stimulation (Fig. 1a). To gain mechanistic insights of PCa 3D genome organization, we further generated H3K27ac and CTCF HiChIP data in CRISPR-Cas9-mediated genomic deletion and MYC-overexpressed 22Rv1 cells. In total, 31 HiChIP libraries, including replicates, were quantified via pair-end (PE) sequencing at an average of 215.5 million (M) ± 94.0 M PE reads per sample.

HiChIP technique identifies RE-associated interactions by antibody-mediated immunoprecipitation on the Hi-C ligated chromatin[13]. Therefore, ChIP peaks-based loop calling is crucial to capture reliable RE-associated interactions. Thus, we first employed the HiCUP-hichipper pipeline to identify ChIP peaks-based loops and then calculated the statistical significance of looping strength by modeling the distribution of mated PE loop reads at all possible anchors (Fig. S1a, b). The HiCUP quality control (QC) reports showed a very high proportion of valid pairs in our HiChIP libraries during the filtering step (Fig. S1c). The de-duplication QC suggested most of the valid pairs were unique di-tags, which are mainly the expected cis-far di-tags (>10Kbp), indicating the high quality of our HiChIP libraries (Fig. S1c). We identified 74,514 high confidence H3K27ac loops (FDR < 0.05) in androgen-deprived VCaP cells, and found the loop numbers dropped to 61,748 and 46,819 after 2 h and 24 h DHT treatment, respectively (Fig. 1b). In sharp contrast, the numbers of AR loops increased from 9751 under androgen-deprivation condition to 24,982 and 24,737 after 2 h and 24 h DHT treatment, respectively (Fig. 1b), suggesting the involvement of activated AR sites in 3D chromatin interaction. Overall, 37.0% of H3K27ac anchors overlap with 50.1% of AR anchors (Fig. S1d), indicating most H3K27ac loops are independent of AR. The opposite effects of androgen on H3K27Ac-specific and AR-specific looping are coincident with AR-mediated cofactor redistribution mechanisms[14].

### H3K27ac and AR loops play distinct roles in androgen-induced gene regulation

To better understand the function of RE-associated interactions in androgen-driven transcription, we integrated H3K27ac and AR HiChIP with RNA-Seq data in VCaP cells with vehicle, 2 h or 24 h DHT treatment. The number of H3K27ac loops with 3-10 mated PE reads showed moderate and pronounced reductions after 2 h and 24 h androgen

treatments, respectively, but the number was stable for H3K27ac loops with more reads (Fig. S1e). Conversely, we observed a marked elevation in the number of all types of AR loops after 2 h and 24 h DHT treatments (Fig. S1f). Consistent with the loop count changes, the average strength of H3K27ac loops anchored at expressed genes was diminished after 24 h DHT treatment ($P < 2.2e{-}16$), while the strength of AR loops was markedly elevated with both 2 h ($P = 3.3e{-}09$) and 24 h ($P = 3.4e{-}09$) DHT treatments (Fig. 1c, S1g and S1h). We next characterized the strength dynamics of loops anchored at differentially expressed genes. For the DHT-upregulated genes, H3K27ac loop strength showed no evident change at 2 h ($P = 0.10$) or 24 h ($P = 0.29$) after DHT treatment, but AR-mediated interactions were markedly reinforced by both 2 h ($P = 1.9e{-}06$) and 24 h ($P = 3.3e{-}06$) DHT treatment (Figs. 1d, 1e, S1h and S1i). For the downregulated genes, H3K27ac loop strength was slightly attenuated by 2 h ($P = 1.3e{-}10$) DHT treatment and further reduced by 24 h ($P = 1.4e{-}13$) DHT treatment (Fig. 1d and S1h). AR-associated interaction dynamics were neglectable at DHT-downregulated genes, especially for genes suppressed by 2 h DHT treatment (Fig. 1e and S1i), suggesting that the direct function of AR in acute androgen stimulation is transcriptional activation, not repression. *IL2ORA* and *CD82* are example genes upregulated by 2 h and 24 h DHT treatments, respectively (Fig. 1f and S2a). In line with their gene expression dynamics, AR-associated enhancer-promoter loops emerged from 2 h at *IL2ORA* gene and 24 h at *CD82* gene after DHT treatment, while H3K27ac loops at the two genes were not significantly changed by DHT treatment (Fig. 1g and S2b). Consistently, quantitative 3 C assay reveals a significant increase of chromatin interactions between *IL2ORA* promoter and enhancers after 24 h DHT treatment (Fig. 1h). On the contrary, H3K27ac loops were diminished from 2 h at *FOXN2* gene and 24 h at *BTF3* gene after DHT treatment, in line with transcriptional repression of these genes (Fig. S2c-f). No AR loops were observed at *FOXN2* and *BTF3* genes, supporting the indirect repressive function of AR (Fig. S2e and S2f). These results indicate AR-associated chromatin contacts are significantly strengthened upon androgen stimulation.

We previously reported redistribution of cofactors mediates androgen-activation of AR-binding sites and repression of H3K27ac-binding enhancers[14]. Similar mechanisms may engage with AR-associated chromatin contacts (enriched at androgen-stimulatory loci) versus the overall H3K27ac-associated chromatin contacts (enriched at androgen-repressive loci). To identify TFs involved in the androgen-induced transcription repression, the DHT-suppressed genes were classified into four categories based on the overlap between gene promoters and H3K27ac/AR loops (Fig. S2g). We reasoned that the H3K27ac loops anchored at DHT-suppressed gene promoters without AR loops were related with other TFs (not AR) functioning in transcription repression. To identify the involved TFs, we compared distal anchors of these H3K27ac loops with TCGA PCa ATAC-Seq peaks, and subjected the overlapping genomic regions to motif enrichment analysis. As expected, AR and FOXA1 motifs were found among the top-ranked motifs enriched in the distal anchors of H3K27ac+/AR+ loops anchored at DHT-activated gene promoters (Fig. S2h, Supplementary data 1). CTCF and KLF4 motifs were significantly enriched in both DHT-activated gene promoter-related distal H3K27ac+/AR+ anchors and DHT-suppressed gene promoter-related distal H3K27ac anchors, indicating their universal function in transcription regulation (Fig. S2h, Supplementary data 1 and 2). Among the top 5 enriched motifs, ERG, NFYA, and ETV4 were specifically associated with H3K27ac loops anchored at DHT-suppressed gene promoters (Fig. S2h, Supplementary data 2), suggesting these TFs play important roles in the regulation of androgen-induced transcription repression. Indeed, ERG was reported to repress AR-mediated transactivation[15]. Together, these analyses pinpointed the potential TFs participating in the AR-independent enhancer-promoter interactions of androgen-repressed genes.

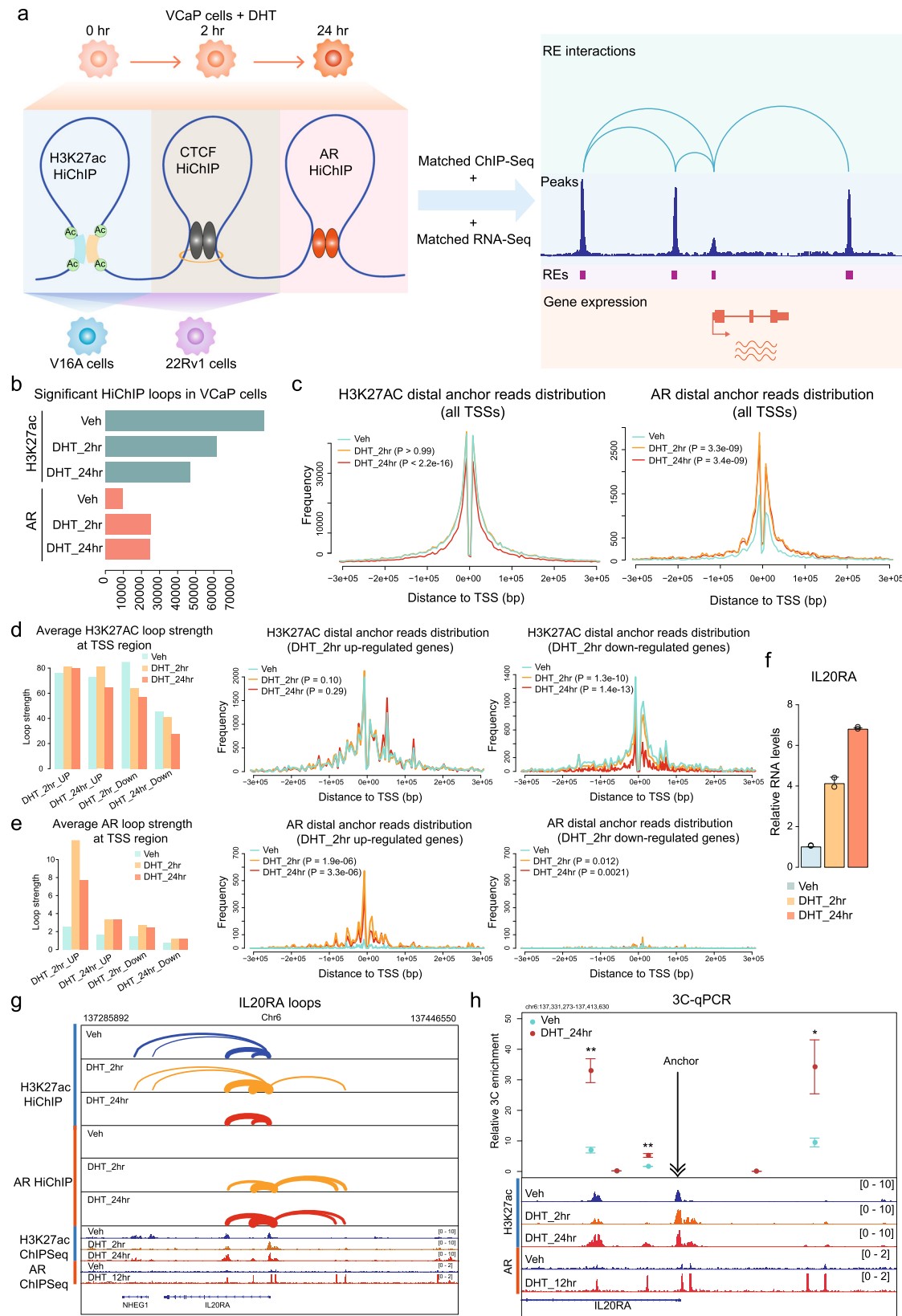

## Cell-type-specific CTCF loops regulate the expression of PCa-related genes

In both AR+/H3K27ac+ and H3K27ac-only loop anchors, CTCF was the top one enriched motif, suggesting a fundamental role of CTCF in the establishment of RE interactions in PCa. To further delineate the function of *CTCF* in PCa, we generated CTCF-mediated chromatin

contact maps by HiChIP in VCaP and 22Rv1, two widely used PCa cell lines derived from a vertebral metastasis of a prostate cancer patient and a human prostatic carcinoma xenograft, respectively. Using merged CTCF ChIP-Seq peaks as anchors, we identified 127,197 and 114,435 CTCF loops in two VCaP replicates, and 159,462 and 164,907 in two 22Rv1 replicates, respectively, which is comparable to the loop

**Fig. 1 | Profiling of regulatory element interactome in PCa cell lines. a** Overview of HiChIP experiments and analyses in PCa cell lines. **b** The number of significant loops (FDR < 0.05) in VCaP HiChIP assays. From top to bottom, $n$ = 74514, 61748, 46819, 9751, 24982, 24737, respectively. **c** Strength of H3K27ac and AR loops anchored at TSSs of all protein-coding genes. The frequency curves showed the normalized read number (loop strength) distribution at anchors distal to TSSs. The strength of each loop was normalized by the number of total 'cis-far' unique valid pairs. $P$-values were determined by paired t-test. For TSSs, $n$ = 19962. **d** Strength of H3K27ac loops anchored at genes downregulated or upregulated by 2 h DHT treatment. The bar plot summarized the strength of loops-anchored gene TSSs. $P$-values were determined by paired t-test. For TSSs of upregulated and down-regulated genes, $n$ = 350 and 205, respectively. **e** Strength of AR loops anchored at

genes downregulated or upregulated by 2 h DHT treatment. The bar plot sum-marized the strength of loops anchored at gene TSSs. $P$-values were determined by paired t-test. For TSSs of upregulated and downregulated genes, $n$ = 350 and 205, respectively. **f** The expression of *IL20RA* was upregulated from the early time point (2 h) after DHT stimulation. $n$ = 2. Data represent means ± SD. **g** For the *IL20RA* gene, AR loops were boosted as early as 2 h after DHT stimulation. **h** 3C–qPCR assay of the *IL20RA* genomic region. The data represents relative frequencies of interaction between the anchor region near the *IL20RA* TSS and selected PstI digestion sites (circles). $n$ = 3. Data represent means ± SD. $P$-values were determined by Student's t-test. *$P$ < 0.05; **$P$ < 0.01. Source data are provided as a Source Data file.

numbers of published CTCF HiChIP data[16,17] (Fig. S3a). The replicates of the same cell line are highly correlated (r = 0.9318 for VCaP replicates; r = 0.9271 for 22Rv1 replicates), while the samples between different cell lines were less correlated (r = 0.6958 and 0.6720), indicating the reliability of these CTCF HiChIP datasets (Fig. 2a–c and S3b). Only 44.92-63.63% of the CTCF HiChIP loops were common between VCaP and 22Rv1 cells (Fig. S3c), while 75-78% of CTCF ChIP-Seq peaks were shared between VCaP and 22Rv1 cells (Fig. S3c), suggesting CTCF HiChIP captures a higher cell-type heterogeneity compared with CTCF ChIP-Seq. The common CTCF loops displayed a higher proportion of strong interactions (more than 10 PE mated reads) than cell-type-specific CTCF loops (Fig. S3d). In line with CTCF HiChIP results, Hi-C interaction signal of common CTCF loops was much stronger than cell-type-specific CTCF loops (Fig. S3e). The higher cell-type diversity of CTCF loops than peaks was further validated by independent CTCF ChIP-Seq and HiChIP data sets of GM12878 and Hela cells (Fig. S3f)[16–18]. The CTCF looping data in GM12878 and Hela also confirmed that common CTCF loops were overall stronger than cell-type-specific CTCF loops (Fig. S3g). The high-level heterogeneity of CTCF looping in PCa cell lines suggests the engagements of 3D chromatin architecture with additional variables.

CTCF insulates RE interactions by forming CTCF-CTCF loops anchored at TAD boundaries, and can also regulate RE interactions by directly binding to active REs. We observed a high ratio of H3K27ac modification in both common and cell-type-specific CTCF loop anchors, and the H3K27ac-positive CTCF anchors contained a higher ratio of promoter regions than H3K27ac-negative CTCF anchors, indicating the importance of CTCF for RE activation (Fig. S3h, S3i). Within common loops, the H3K27ac-marked CTCF loops had the highest strength (25 ± 31 PE mated reads) and shortest distance (125.3 ± 163.6 Kb; Fig. 2d). For both common and cell-type-specific CTCF loops, the H3K27ac-marked CTCF loops encompassed more genes than other CTCF loops (Fig. 2e). Compared with H3K27ac-negative (H3K27ac −/−) CTCF loops, CTCF loops with double-positive H3K27ac in two anchors (H3K27ac +/+) were positively related with cell-type-specific gene expression in both 22Rv1 ($P$ = 1.3e-08) and VCaP cells ($P$ < 2.2e-16; Fig. 2f), suggesting H3K27ac +/+ CTCF loops play a role in promoting gene transcription. The intermediate expression alteration of genes within H3K27ac -/+ CTCF loops compared with H3K27ac −/− and +/+ CTCF loops indicate both transactivation and insulation function of H3K27ac -/+ CTCF loops (Fig. 2f). To find the biological relevance of CTCF loops, we conducted pathway enrichment analysis using genes annotated to CTCF anchors. Many cancer-related pathways were enriched in H3K27ac -/+ common CTCF loop anchors, such as "Pancreatic cancer" and "Acute myeloid leukemia" (Fig. 2g). Interestingly, "Prostate cancer" pathway genes were enriched in both common and cell-type-specific CTCF loop anchors (Fig. 2g), which is consistent with the tissue origin of VCaP and 22Rv1 cells, suggesting CTCF looping is involved in tissue-specific gene regulation.

An example of CTCF-loop-regulated gene is *TMC5*, which is transcriptionally suppressed by androgen stimulation in VCaP cells (Fig. S3j) and has been reported to promote prostate cancer cell

proliferation[19]. A 22Rv1-specific H3K27ac-negative CTCF peak, which is at -10 Kb upstream of *TMC5* promoter, was linked to the H3K27ac-positive CTCF peak at *TMC5* promoter, resulting in the insulation of *TMC5* promoter from upstream enhancers in 22Rv1 cells (Fig. 2h). This CTCF site and loop were not observed in VCaP cells, and accordingly, upstream enhancers were connected to *TMC5* promoter by H3K27ac loops (Fig. 2h). In line with the chromatin looping, the expression of *TMC5* is much higher in VCaP cells than in 22Rv1 cells (Fig. 2h). Consistent with the previous report that cell-type-specific CTCF occupancy is negatively correlated with DNA methylation level[7], the CTCF binding affinities showed significant negative correlations with the methylation levels of all the 8 CpGs at this site in ENCODE cell lines (Fig. 2i). Importantly, we observed a positive correlation between DNA methylation of this CTCF site and *TMC5* expression levels in PCa clinical samples (Fig. 2j), which emphasizes the clinical significance of the insulation function of this CTCF loop.

**Deletion of a CTCF site near the *MYC* promoter leads to profound changes of CTCF looping**

After linking the CTCF loops to the regulation of PCa-related genes, we sought to obtain a global view of the CTCF-mediated contacts between CTCF binding sites and essential cancer genes. We retrieved 2,134 essential genes in cancer development from DepMap[20], and divided the CTCF sites looping to those genes into H3K27ac-negative (H3K27ac-) and H3K27ac-positive (H3K27ac+) groups. Then, the fold changes of CTCF binding affinities (22Rv1 vs. VCaP) were compared with expression changes of matched essential genes. The CTCF sites, of which the binding affinity changes are consistent with expression changes of matched essential genes, are three times the number of CTCF sites with opposite trends ($n$ = 676 and 221; Fig. 3a). Of note, while the number of H3K27ac- CTCF sites is comparable to that of H3K27ac+ CTCF sites in the consistent trend group ($n$ = 343 and 333), for the CTCF sites with opposite trends, H3K27ac- site number is much higher than H3K27ac+ site number ($n$ = 134 and 87; $P$ = 0.013; Fig. 3a). These results suggest the H3K27ac- CTCF sites are more likely to suppress their looped essential genes compared with H3K27ac+ CTCF sites.

Among the H3K27ac- CTCF sites, one site at -10 Kb upstream of the MYC gene (hg19, chr8:128737774-128738489; referred to as "−10Kb CTCF site" hereafter) was looped to *MYC* promoter, and the CTCF binding affinities at this site are negatively correlated with *MYC* expression levels in PCa cell lines (Fig. 3a, b). Consistently, our group recently found the deletion of the −10Kb CTCF site led to a dramatic upregulation of *MYC* expression[1]. In PCa, the *MYC*-located 8q24 region contains lots of CTCF-mediated short and long chromatin loops, but the function of these CTCF loops is largely unknown (Fig. S4a). To gain a deeper insight into the −10Kb CTCF site function in 3D chromatin interaction, we generated CTCF and H3K27ac HiChIP data with control (sgCtrl) and −10Kb CTCF site deletion (sgDele-10kb) 22Rv1 cells (Fig. 3b and S4b). As shown by CTCF HiChIP, the −10Kb CTCF site looped to another CTCF site at *MYC* promoter (Fig. 3b), which was reported as the enhancer-docking site for *MYC* transcription[2]. As expected, the deletion of "−10Kb CTCF site" completely disrupted the

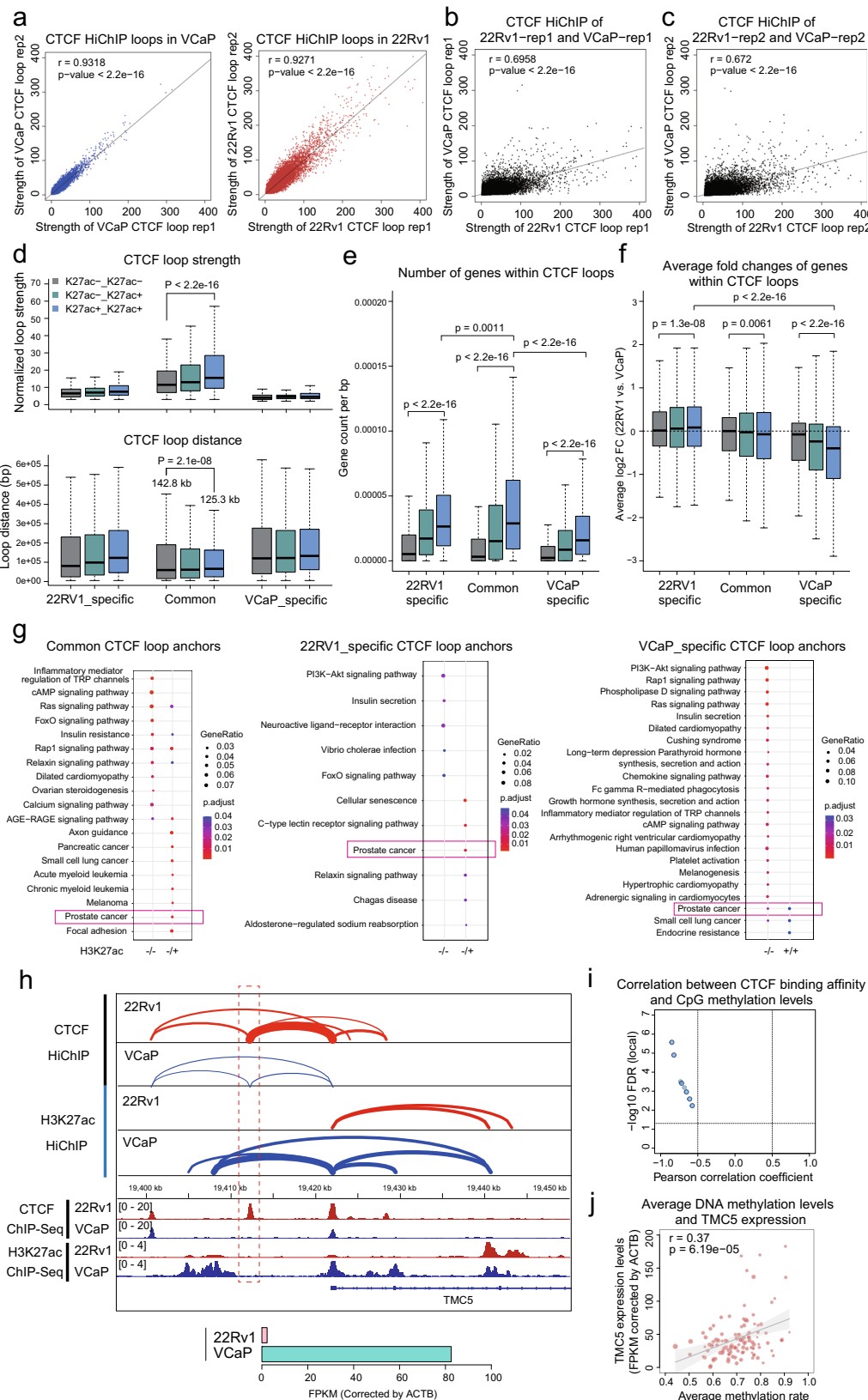

CTCF loop between −10Kb CTCF site and *MYC* enhancer-docking site, and introduced extensive new H3K27ac-associated loops connecting multiple upstream enhancers to *MYC* region (Fig. 3b, S4c and S4d). We and others previously reported that a cluster of super-enhancers within PCAT1/2 region interact with *MYC* promoter to regulate *MYC* expression in VCaP cells[14,21,22]. Here, we found the super-enhancers

within PCAT1/2 region were also robustly looped to *MYC* in 22Rv1 cells, but the looping between them was not significantly changed by −10Kb CTCF site deletion (Fig. S4d). Interestingly, the H3K27ac-associated interaction between super-enhancers of the PCAT1/2 locus was weakened by −10Kb CTCF site deletion (Fig. 3b), indicating a shift of interacting co-factors to *MYC*-related loops upon *MYC* upregulation.

**Fig. 2 | Cell-type-specific CTCF looping regulates PCa-associated genes.**
**a** Scatter plots showing the association of CTCF loop strength between two repli-cates in VCaP and 22Rv1, respectively. $n = 167054$ and $218739$ from left to right, respectively. *P*-values were calculated using Pearson correlation. **b** Scatter plots showing the association of CTCF loop strength between VCaP replicate #1 and 22Rv1 replicate #1. $n = 217761$. *P*-value was calculated using Pearson correlation. **c** Scatter plots showing the association of CTCF loop strength between VCaP replicate #2 and 22Rv1 replicate #2. $n = 210816$. P-value was calculated using Pearson correlation. **d** Boxplots showing the strength and distances of CTCF loops classified based on cell-type specificity and H3K27ac status of two anchors from the same loop. Box plots indicating the mean (middle line), 25th and 75th percentile (box) and 10th and 90th percentile (whiskers), and $n = 24661$, $24708$, $8804$, $17267$, $21223$, $8967$, $12496$, $11277$, $3348$ for boxes from left to right, respectively. *P*-values were calculated by Wilcoxon signed-rank test. **e** The number of genes within CTCF loops. The gene numbers were normalized by CTCF loop distances. Box plots indicating the mean (middle line), 25th and 75th percentile (box) and 10th and 90th percentile (whiskers), and $n = 24661$, $24708$, $8804$, $17267$, $21223$, $8967$, $12496$, $11277$,

3348 for boxes from left to right, respectively. *P*-values were calculated by Wil-coxon signed-rank test. **f** Average log2 expression fold changes (22Rv1 vs. VCaP) of genes within CTCF loops. Box plots indicating the mean (middle line), 25th and 75th percentile (box), and 10th and 90th percentile (whiskers), and $n = 24661$, $24708$, $8804$, $17267$, $21223$, $8967$, $12496$, $11277$, $3348$ for boxes from left to right, respec-tively. *P*-values were calculated by Wilcoxon signed-rank test. **g** KEGG pathway enrichment analysis of genes annotated to the anchors of CTCF loops. The proxi-mity of genes to anchors is restricted to 3Kb. **h** Upper: A cell-type-specific CTCF binding insulates the interaction between a distal enhancer and *TMC5* promoter by forming CTCF-CTCF loops. The 22Rv1-specific CTCF binding was highlighted within a red dashed rectangle. Bottom: *TMC5* expression levels in VCaP and 22Rv1 cells based on RNA-Seq data. **i** Pearson correlation coefficients between DNA methyla-tion ratio of CpGs and CTCF binding affinity at this CTCF site in 20 ENCODE cell lines. Each circle denotes a CpG within this CTCF site. **j** Scatter plot showing a positive correlation between average DNA methylation levels at this CTCF site and *TMC5* expression levels in Changhai 2020 data set. The error band indicates SEM. *P*-value was calculated using Pearson correlation.

Together, these data suggest that a subset of H3K27ac- CTCF sites can block enhancer-promoter interaction by forming CTCF-CTCF loop to gene promoter.

## MYC is enriched in CTCF loop anchors
We then examined the impact of the −10Kb CTCF site deletion on chromatin architecture at genome-wide scale. Unexpectedly, besides the local chromatin architecture alteration, we also observed a global impact of this CTCF site deletion on RE interactions. Differential looping analysis identified 261 and 2,952 significantly changed CTCF and H3K27ac loops, respectively, across the whole genome. Upon −10Kb CTCF site deletion, there were 13 chromosomes with more upregulated CTCF loops, 6 chromosomes with an equal number of upregulated and downregulated CTCF loops, and only 4 chromosomes with more downregulated CTCF loops (Fig. 3c). Since the primary effect of −10Kb CTCF site deletion is to drive MYC upregulation, we speculated the global alteration of CTCF loops is associated with chromatin occupation of MYC. In agreement with our hypothesis, MYC binding sites were enriched in both sgDele-10Kb-specific and sgCtrl-specific CTCF anchors (Figs. S4e, 3d). To validate the activation of MYC gene after repression of −10Kb CTCF site, we then targeted this site using dCas9-KRAB complex. Repression of −10Kb CTCF site resulted in the decrease of CTCF binding at this site and enhancement of MYC expression, cell proliferation, MYC binding affinity at target genes and MYC target gene expression (Fig. S4f–g), supporting an increasement of global MYC activity.

To further determine the molecular basis of MYC function at CTCF anchors, we performed motif enrichment analysis using CTCF binding peaks within CTCF anchors. The MYC motif was recognized as one of the mst enriched motifs (rank 3, $P = 1e-23$) in MYC+/H3K27ac- CTCF anchors, and showed moderate enrichment in MYC+/H3K27ac+ CTCF anchors (rank 12, $P = 1e-6$; Fig. 3e). Motif distribution scanning by Homer confirmed the highest enrichment of MYC motif in MYC +/H3K27ac- CTCF anchors, and revealed the colocalization of MYC motif and CTCF motif (Fig. 3f). Importantly, the colocalization pattern of MYC and CTCF was supported by cistrome data. MYC occupation was generally accompanied by higher enhancer activity and chromatin accessibility as revealed by 22Rv1 H3K27ac ChIP-Seq and PCa ATAC-Seq data, but also indicated higher CTCF binding in 22Rv1 cells and cohesin binding in A549 cells at H3K27ac- CTCF sites (Fig. 3g and S4k). These results suggest MYC and CTCF motifs are co-localized at a subset of genomic regulatory elements, which is associated with the co-occupancy of MYC and CTCF at these sites.

## MYC facilitates CTCF chromatin binding
To further assess the effects of *MYC* on CTCF-mediated 3D chromatin organization, we performed RNA-Seq, MYC/CTCF/H3K27ac ChIP-Seq

and CTCF/H3K27ac HiChIP assays in control and *MYC*-overexpressed 22Rv1 cells (Figs. 4a, 4b). MYC overexpression induced 8,998 new MYC peaks (54.2% of total MYC peaks) in 22Rv1 cells (Fig. 4c), and highly increased the binding affinities of MYC peaks (Fig. 4d). After MYC overexpression, there were 4,045 emerged and 6,676 diminished CTCF peaks (Fig. 4c), respectively, which accounted for a small fraction of total CTCF peaks (5.6% and 9.2%). In MYC-overexpressed cells, 54.1% of MYC peaks overlapped with 14.1% of CTCF peaks (Fig. 4e and S5a), showing a disproportional MYC association with CTCF on the chro-matin. MYC overexpression increased CTCF occupancy at CTCF/MYC common sites, especially at H3K27ac- sites (Fig. 4f), suggesting a mechanistic link of MYC to gene repression. In addition, a genome-wide redistribution of H3K27ac signal was induced by MYC over-expression, with 10,817 emerged and 14,327 diminished H3K27ac peaks (Fig. 4c). H3K27ac levels were slightly repressed by MYC at both MYC+ or MYC- regions (Fig. 4g). We also observed the chromatin colocalization of CTCF and MYC in LNCaP and VCaP ChIP-Seq data (Fig. S5b, S5c). Although the portions of overlapping peaks in LNCaP and VCaP cells were smaller than that in 22Rv1 cells, the MYC+ CTCF binding was still stronger than MYC- CTCF binding in LNCaP and VCaP cells, supporting that MYC facilitates CTCF chromatin occupancy (Fig. S5d).

Next, we examined whether there is a physical interaction between MYC and CTCF proteins. By co-immunoprecipitation (co-IP), we observed the interaction between MYC and CTCF in both control and MYC-overexpressed 22Rv1 cell (Fig. 4h). Similarly, co-IP assay also showed the interaction of CTCF and MYC proteins in V16A cells (Fig. S5e). The MYC-CTCF interaction was further confirmed by GST-pulldown assay, co-IP assay of tagged proteins, and proximity ligation assay (Fig. 4i–k). Taken together, our data suggest MYC interacts with CTCF and facilitates its chromatin occupancy at CTCF/MYC common sites.

## MYC represses a subset of neuroendocrine lineage plasticity genes by enhancing CTCF-mediated chromatin looping
After confirming the effect of MYC on CTCF binding, we next investi-gated the impact of MYC on CTCF-mediated looping and gene expression. Differential expression analysis (FDR < 0.05 and fold change > 2; DESeq2) showed a higher number of repressed ($n = 479$) compared with activated ($n = 181$) genes upon MYC overexpression (Fig. 5a). While MYC-activated genes were enriched cell proliferation gene sets, such as "E2F_TARGETS" and "mesenchymal cell prolifera-tion", MYC-repressed genes were enriched in neurogenesis and endocrine-related gene sets like "regulation of neurogenesis" and "insulin secretion" (Fig. 5b and S6a–c). Since neuroendocrine trans-differentiation plays an important role in PCa progression, we then assessed whether the pan-neuroendocrine tumor (pan-NET) genes that

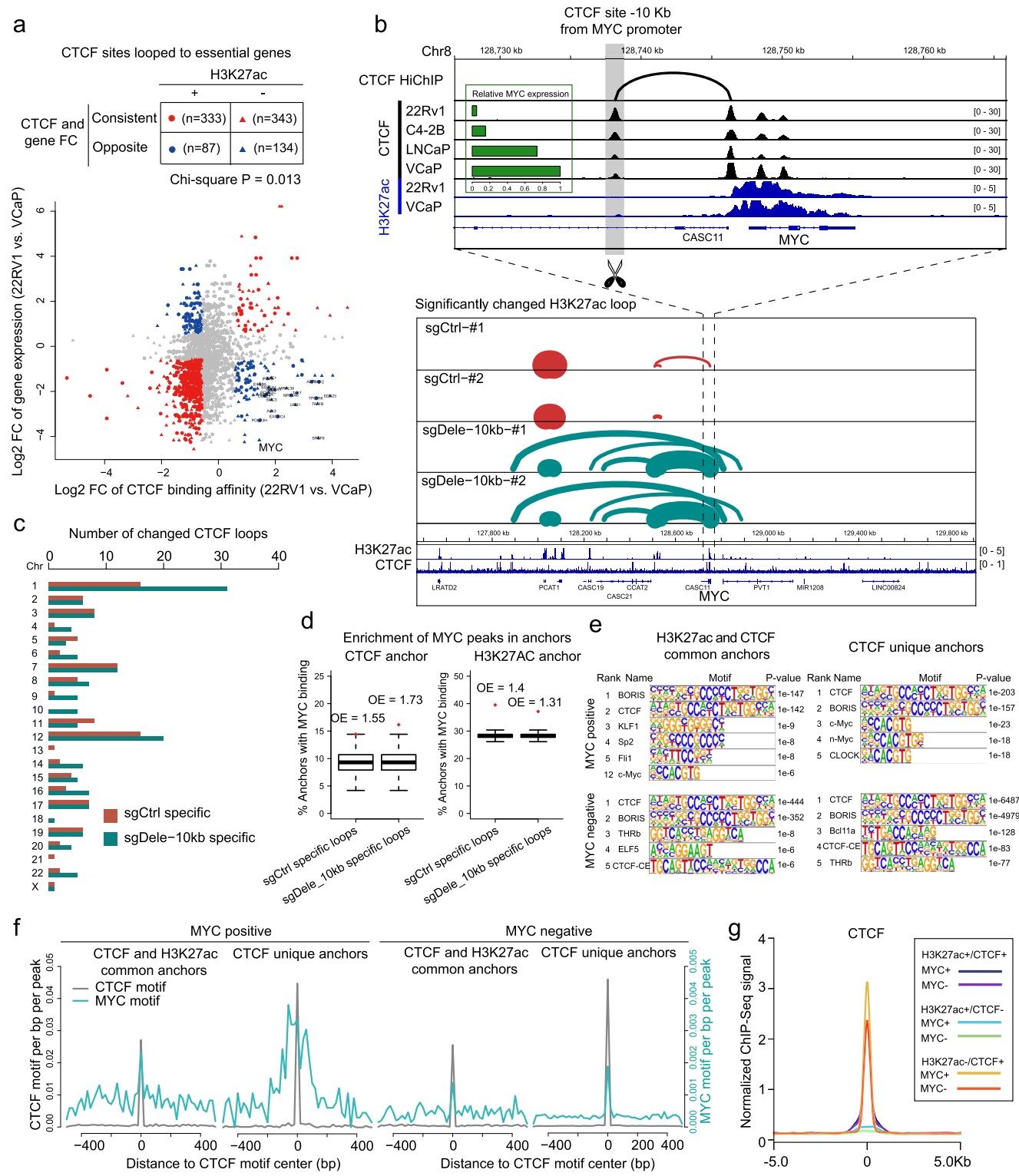

we previously defined[23] were regulated by MYC. Gene Set Enrichment Analysis (GSEA) showed a dramatical downregulation of pan-NET genes by MYC (*P* = 5.19e-06; Fig. 5c). Of 88 pan-NET genes, 51 were significantly repressed by MYC (FDR < 0.05), including canonical neuroendocrine marker gene CHGA and CHGB (Fig. S6d). Consistent with gene expression alteration, H3K27ac-associated chromatin looping was enhanced at the promoters of MYC-upregulated genes and diminished at promoters of MYC-downregulated genes (Fig. 5d and S6e). In line with its stimulatory effects on CTCF binding, MYC expression caused more strengthened than weakened CTCF loops

(*n* = 5609 and *n* = 2502, respectively). The distribution curves of both up- and downregulated CTCF loops show peaks at 150-330 Kb (Fig. 5e). A large fraction of enhanced CTCF loops spanned TAD boundaries (Fig. 5f and S6f), indicating their long-range insulation function.

We then sought to connect the CTCF looping changes to the MYC-induced gene dysregulation by analyzing the crossover between upregulated CTCF loops and downregulated H3K27ac loops (Fig. 5g). Our analysis identified 941 downregulated H3K27ac loops crossover with upregulated CTCF loops, and those H3K27ac-associated interactions were potentially insulated by CTCF. Those H3K27ac loops were

**Fig. 3 | Deletion of a CTCF site near the MYC promoter leads to re-organization of CTCF looping. a** The scatter plot showing the fold changes of CTCF binding affinities at CTCF sites looping to gene promoters and the expression fold changes of corresponding genes. The interactions between CTCF sites and gene promoters were determined by CTCF HiChIP loops. VCaP and 22Rv1 CTCF ChIP-Seq data were obtained from ENCODE and gene expression data were retrieved from GSE25183. The CTCF-gene pairs with consistent binding/expression fold changes and both absolute fold changes > 1.5 are labelled red, and opposite fold changes are labelled blue. The dots and triangles indicate CTCF sites with and without H3K27ac overlapping, respectively. Genes in representative CTCF-gene pairs were annotated. $n = 2088$. *P*-value was calculated by Chi-square test. **b** Upper: The highlighted CTCF site was connected to *MYC* promoter by a CTCF loop. The CTCF binding affinities at this site (−10 Kb from *MYC* promoter) were negatively correlated with *MYC* expression in prostate cancer cell lines. The CTCF site −10 Kb from *MYC* promoter was deleted by CRISPR/Cas9-mediated knock-out in 22RV1 cells. The control (sgCtrl) and CTCF deletion (sgDele-10Kb) cells were then used for H3K27ac and CTCF HiChIP experiments. Bottom: Significantly changed H3K27ac loops at MYC region by the "−10 Kb CTCF site" deletion. For each group of H3K27ac HiChIP, two

biological replicates were performed. **c** Number of significantly changed CTCF loops at each chromosome before and after "−10Kb CTCF site" deletion. **d** Enrichment analysis of MYC binding at the anchors of dysregulated CTCF or H3K27ac loops. Orange points represent the actual ratio of dysregulated loop anchors with MYC binding. Each box represents 500-time random sampling from all CTCF or H3K27ac loop anchors. Same number of loop anchors as in the sgCtrl-specific or sgDele-10Kb-specific anchor set was used for random sampling, respectively. O/E (observed vs. expected) was calculated by comparing the overlap percentage of actual dysregulated loop anchors with that of the average of randomly sampled anchors. Box plots indicating the mean (middle line), 25th and 75th percentile (box), and 10th and 90th percentile (whiskers). Points were highlighted by red if $P < 0.05$. *P*-values were determined by Student's t-test. $n = 215, 297, 2896, 2322$ from left to right, respectively. **e** Motifs enriched in the CTCF peaks at CTCF anchors of indicated loops. **f** Motif distribution in CTCF peaks at CTCF anchors of indicated loops. For each CTCF peak, the location of the best-ranked CTCF motif was used as the center for the motif density plot. **g** The aggregated CTCF ChIP-Seq signal in 22Rv1 cells at indicated peaks.

mapped to 4,557 gene promoters, of which MYC-downregulated genes accounted for a significant higher portion compared with MYC-upregulated genes ($P = 0.0042$; Chi-squared test; Fig. 5h). We then took *CDK5R2*, one of MYC-repressed pan-NET genes, to exemplify the CTCF effect. A robust H3K27ac loop connected a distal enhancer to *CDK5R2* promoter, potentially maintaining *CDK5R2* expression (Fig. 5i). MYC overexpression facilitated CTCF binding in this region and thus introduced new CTCF loops intersecting with the H3K27ac loop of *CDK5R2*, resulting in the weakening of promoter-enhancer interaction of *CDK5R2* gene (Fig. 5i). To assess the contribution of CTCF sites in MYC-repressed *CDK5R2* expression, we targeted three CTCF sites at the anchors of new CTCF loops by dCas9-KRAB strategy. CTCF binding affinity was significantly reduced at all three sites by CRISPRi (Fig. 5k). Consequently, *CDK5R2* mRNA levels significantly increased in all three CRISPRi cell lines under MYC overexpression conditions (Fig. 5k).

Finally, we checked the association between MYC activities and pan-NET scores in several PCa clinical data sets. High pan-NET scores were significantly related to low MYC activities in both primary PCa and CRPC data sets (Fig. 5j and S6g), highlighting the clinical significance of MYC-induced repression of neuroendocrine genes. Overall, our data suggest MYC suppresses neuroendocrine gene transcription by enhancing CTCF-mediated chromatin looping.

## Discussion

As a key architectural protein, CTCF bridges the genome topology and gene expression regulation. By exploring the CTCF-associated chromatin contact map in PCa, we found a direct association between selective CTCF looping and gene expression regulation. For both *TMC5* and *MYC* genes, the cell-type-specific CTCF binding sites near PCa-related genes form CTCF-CTCF loops and interrupt the access of enhancers to gene promoters. This observation is consistent with a previous integrative analysis of Hi-C and CTCF ChIP-Seq data, which showed that several CTCF sites near gene promoters inhibited gene expression in PCa[24]. Furthermore, the methylation level at a cell-type-specific CTCF site looping to *TMC5* gene is positively correlated with *TMC5* expression in a clinical PCa cohort, indicating the potential implication of context-dependent CTCF sites in the development of DNA methylation-based PCa biomarkers.

*MYC* is known to reside in an enhancer-less locus, and its expression was predominantly subjected to regulation by long-range chromatin interaction dynamics. In breast cancer, *MYC* and *PVT1* promoters compete for a cluster of downstream enhancers[3]. In a B-ALL cell line, CTCF regulates *MYC* expression by maintaining the chromatin interaction between *MYC* promoter and a distal downstream enhancer cluster[25]. In PCa, our group reported androgen represses *MYC* transcription by disrupting the interaction between super-enhancers

within *PCAT1* region and *MYC* promoter[14]. However, the effect and mechanism of *MYC* in 3D genome organization was poorly defined. We reported in this study that MYC potentiates CTCF-mediated chromatin looping to suppress the expression of a subset of genes in PCa (Fig. S5h). This finding is also supported by a previous report that *Myc* deletion in activated mouse B cells markedly reduces loop contacts and Rad21 binding at loop anchors[5]. Interestingly, a recent study reported MYC activation strengthens chromatin interactions at super-enhancers and MYC binding sites in U2OS osteosarcoma cells[6]. Together, these findings start to corroborate *MYC* functions in 3D genome. Although our data suggest MYC assists CTCF chromatin binding at MYC/CTCF common sites, we could not exclude additional function of MYC in the regulation of CTCF chromatin occupancy. For example, considering CTCF chromatin binding was reported to display a cell cycle stage-dependent dynamics[26], and MYC is a well-established regulator in cell cycle control[27], MYC-regulated cell cycle progression may also play a role in CTCF binding changes. In addition, as our results were obtained by population-based methodologies, the increase of CTCF chromatin occupancy could rely on either the enhanced CTCF binding affinity or prolonged chromatin residence time of CTCF. Further single-cell epigenomic assays and single-molecule imaging assays are needed to unravel the detailed mechanism of MYC-facilitated CTCF chromatin occupancy.

AR acts mainly as a transcriptional activator but is also involved in the regulation of androgen-induced transcriptional repression[28,29]. Our AR HiChIP showed a prominent increase in AR-associated enhancer-promoter interactions after androgen stimulation, supporting the direct transcriptional activation function of AR from the perspective of 3D genomics. Although AR binding has been reported to be related to several androgen-repressed genes[28,30], our HiChIP analysis of AR loops before and after androgen stimulation decoupled the AR-associated enhancer-promoter interactions from androgen-induced transcription repression. Instead, the dynamics of H3K27ac-associated promoter-enhancer contacts synchronize with gene expression alteration at androgen-repressed genes, consistent with their dependence on TFs other than AR. Our motif analysis identified the ERG motif was enriched in H3K27ac+/AR- enhancers of androgen-repressed genes. The *TMPRSS2-ERG* fusion, which results in *ERG* overexpression, occurs in ~50% of prostate tumors[31,32]. As an oncogenic gene in prostate cancer, ERG regulates proliferation and invasion genes by orchestrating higher-order chromatin organization, which is distinct from AR-associated chromatin connectivity[33–35]. Indeed, ERG has been demonstrated to repress AR-mediated transactivation[15]. In line with these reports, our results further suggest *ERG* may play an important role in the construction of promoter-enhancer looping of androgen-repressed genes.

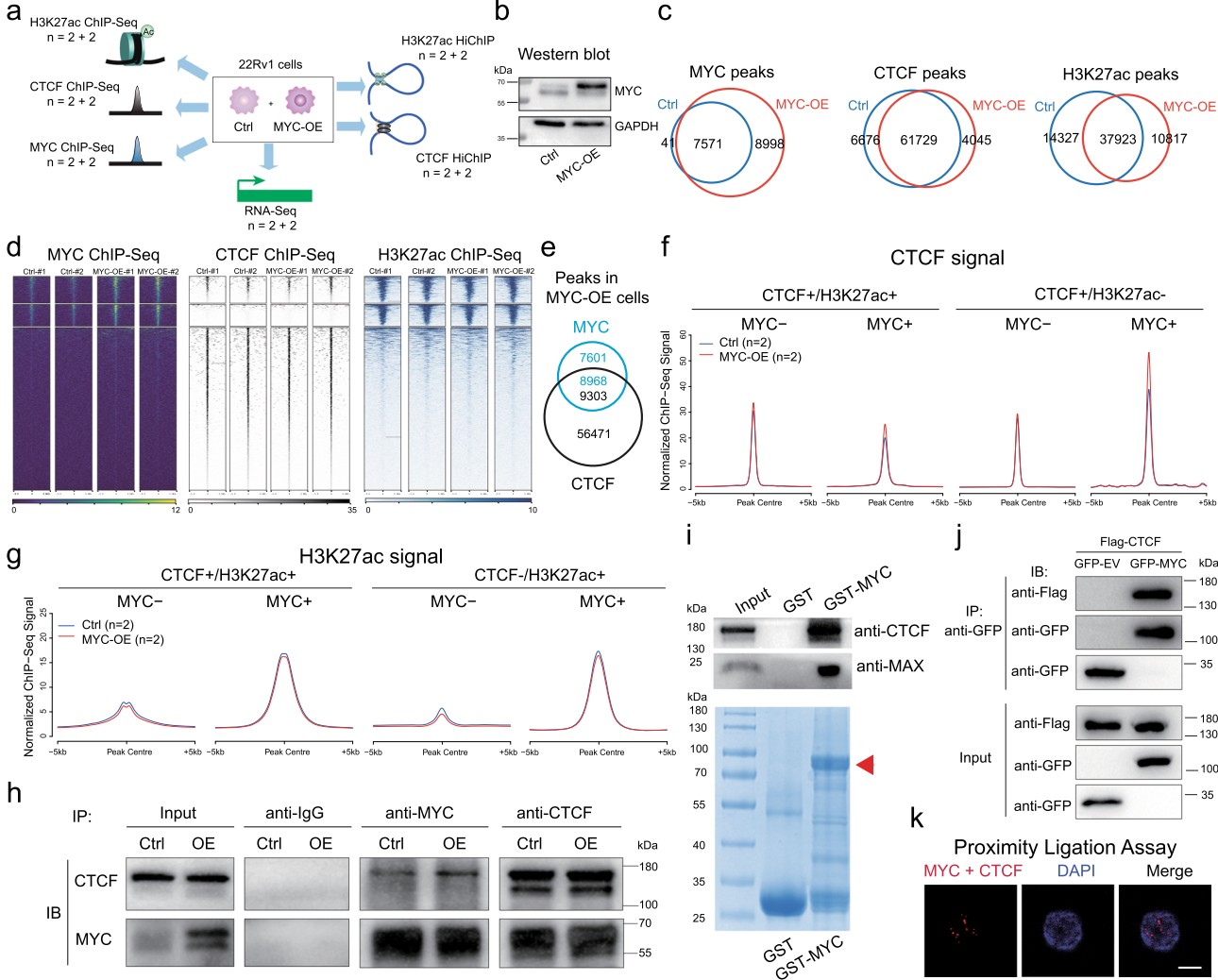

**Fig. 4 | MYC facilitates CTCF chromatin binding. a** Multi-omics assays to evaluate the function of MYC on CTCF binding and looping in 22Rv1 cells. **b** Western blot assay showing the efficiency of MYC overexpression. **c** The overlap of MYC, CTCF and H3K27ac peaks between control (Ctrl) and MYC overexpression (MYC-OE) cells. **d** Heatmaps showing the ChIP-Seq signal of MYC, CTCF and H3K27ac at MYC and CTCF peaks. From top to bottom, MYC and CTCF peaks were separated into shared, MYC-only, and CTCF-only groups. **e** The overlap between MYC and CTCF peaks in MYC-OE cells. **f** The aggregated CTCF ChIP-Seq signal in Ctrl and MYC-OE cells. CTCF peaks were divided into four groups based on MYC and H3K27ac status. **g** The aggregated H3K27ac ChIP-Seq signal in Ctrl and MYC-OE cells. H3K27ac peaks were divided into four groups based on MYC and CTCF status. **h** Co-immunoprecipitation to detect the protein-protein interaction between CTCF and MYC in Ctrl and MYC-OE cells. **i** Western blot and corresponding Coomassie blue staining of GST pull-down assay. 22Rv1 cell lysates were subjected to pulldowns with immobilized GST only or recombinant GST-CTCF protein. Bound protein was probed with anti-CTCF and anti-MAX antibodies by Western blot. The red arrow indicates the GST-MYC band. **j** 22Rv1 cells expressing GFP-MYC and Flag-CTCF were used in co-IP assay. The input and IPed proteins were analyzed by Western blot with anti-GFP and anti-Flag antibodies. **k** Proximity Ligation Assay to detect the in-situ interaction between MYC and CTCF proteins. Nuclei were stained with DAPI. Scale bar, 10 μm. For **b**, **h**, **i**, **j**, and **k**, these experiments were repeated independently three times with similar results. Source data are provided as a Source Data file.

In summary, by interrogating PCa interactome data, we revealed the role of MYC in regulating 3D genome organization. Moreover, we provided a comprehensive collection of 3D-epigenome data sets in multiple PCa cellular models, which would be a valuable resource for PCa genomic research.

## Methods

### Cell lines

All cell lines were cultured at 37 °C with 5% $CO_2$. The medium used for VCaP (CRL-2876, ATCC), 22RV1 (CRL-2505, ATCC), V16A (established by Dr. Amina Zoubeidi's laboratory), and HEK293FT (R70007, Thermo Fisher Scientific) cell culture was supplemented with 10% FBS (GIBCO, 10437-028), 1% streptomycin and 1% penicillin. For androgen stimulation treatment, cells were grown to 50%-60% confluence in a medium containing 5% charcoal-dextran stripped FBS (CDS) for 48 h and then treated by 10 nM DHT for 2 or 24 h. All cell lines used in this study were tested negative for mycoplasma contamination. We utilized ATCC services following extended passages to authenticate by utilizing Short Tanden Repeat (STR) profiling.

### Lenti-viral vector construction and transfection

The *MYC* CDS sequence was cloned into the pLVX-IRES-Puro vector (named pLVX-MYC), and then verified by DNA sequencing. The primer sequences for MYC amplification were listed in Supplementary table 1. The plasmids pLVX-MYC or the negative control (pLVX-NC) were co-transfected with the plasmids of pLP1, pLP2, and pLP/VSVG using the transfection reagent Lipofectamine™ 2000 (Invitrogen, 11668-019) according to the manufacturer's

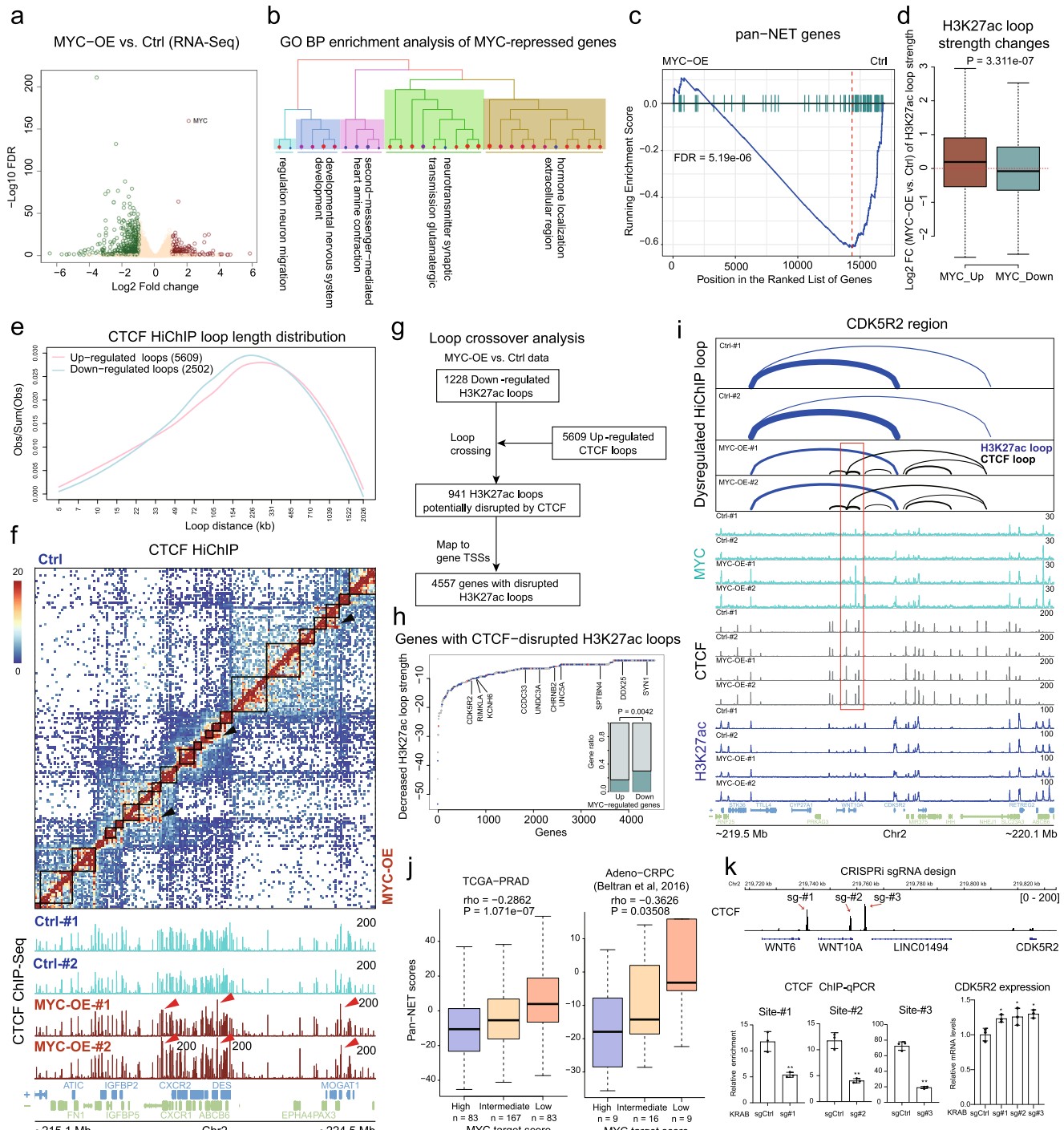

**Fig. 5 | MYC represses neuroendocrine genes by promoting CTCF looping.**
**a** Volcano plot showing dysregulated genes by *MYC* overexpression. *n* = 2. **b** GO BP enrichment analysis of 479 MYC-repressed genes. The GO terms with high similarity were clustered together and summarized. *n* = 2. **c** GSEA plot showing the pan-NET genes were overall suppressed by *MYC* overexpression. **d** Boxplots showing H3K27ac loop strength fold changes at the promoters of upregulated or downregulated genes after *MYC* overexpression. Box plots indicating the mean (middle line), 25th and 75th percentile (box) and 10th and 90th percentile (whiskers). *n* = 854, 5024 from left to right, respectively. P-value was determined by Student's t-test. **e** The length distribution of CTCF loops dysregulated by MYC. **f** Normalized CTCF HiChIP contact matrices of Ctrl and MYC-OE cells at a genomic region of chromosome 2. The black rectangles indicate TAD structures. Black arrows highlight the intra-TAD CTCF looping enhanced by MYC. Red arrows highlight the corresponding CTCF binding increased by MYC. **g** The analysis workflow to identify genes with H3K27ac loops disrupted by increased CTCF looping after *MYC* overexpression. **h** Ranked dot plot showing the decreased H3K27ac loop strength

(MYC-OE - Ctrl) at promoters of 4557 genes from the analysis in (**g**). Red and blue circles indicate genes up- and downregulated by MYC, respectively. The annotated circles were downregulated pan-NET genes. Bar plot showing the overlapping between MYC-dysregulated genes and the 4557 genes. The *P*-value was determined by Chi-square test. **i** Dysregulated CTCF and H3K27ac loops at the genomic region spanning the *CDK5R2* gene in Ctrl and MYC-OE cells. MYC-induced CTCF binding and looping were highlighted. **j** Gene set z-scores of MYC targets and pan-NET genes were negatively correlated in two PCa RNA-Seq data sets. Spearman rho and P-value were shown. Box plots indicating the mean (middle line), 25th and 75th percentile (box) and 10th and 90th percentile (whiskers). *n* = 83, 167, 83, 9, 16, 9 from left to right, respectively. **k** Top: Three CRISPRi sgRNAs were designed to target the three CTCF sites upstream of *CDK5R2* gene, respectively. Bottom: CTCF ChIP-qPCR and RT-qPCR with or without CRISPRi in *MYC* overexpressed 22Rv1 cells. *n* = 3. Data represent means ± SD. P values were two-sided Student's t test. *P < 0.05; **P < 0.01. Source data are provided as a Source Data file.

protocol. The stably-transfected 22Rv1 cells were selected by puromycin (600 ng/mL).

## Western blotting

Cells were harvested and lysed on ice using RIPA buffer (Beyotime, P0013B) containing 1% PMSF (Solarbio, P0100). The proteins were purified by centrifugation (12,000 g at 4 °C for 20 min) and quantified by Bicinchoninic Acid (BCA) protein assay. Total protein concentrations were normalized in all samples. The proteins were heated at 95 °C for 10 min in the loading buffer (Beyotime, P0015L), then resolved in 10% dodecyl sulfate (SDS)-polyacrylamide electrophoresis gel, electrophoresed with Tris-glycine running buffer at 15 V/cm for 1 h, and finally transferred to a polyvinylidene difluoride (PVDF) membrane (Millipore, IPVH00010). This membrane was incubated at room temperature in blocking buffer (5% non-fat dry milk in TBST) for 2 h. Subsequently, the membranes were incubated with primary antibodies rabbit anti-MYC (1:1000; Abcam, ab32072, Rabbit monoclonal [Y69], Lot: GR3377350-5) and rabbit anti-GADPH (1:1000; Cell Signaling Technology, 5174 S, Rabbit monoclonal [D16H11], Lot: 8) at 4 °C overnight. GADPH was used as internal control. The membranes were washed three times with TBST and incubated with horseradish-peroxidase (HRP)-coupled secondary antibody anti-rabbit IgG (1:10,000; ZSbio, zb-2301) for 1 h at room temperature. After the second round of wash, the ECL reagent (Boster, AR1191) was used to visualize protein bands.

## CRISPR/Cas9-mediated −10 Kb CTCF site deletion

22Rv1 clones with the "−10Kb CTCF site" deletion was reported in our previous study[1]. These clones were used for H3K27ac and CTCF HiChIP assays.

## CRISPRi assay

To stably express dCas9-KRAB in 22Rv1 cells, the lentiviral packaging system including the Lenti-dCas9-KRAB-blast plasmid (Addgene plasmid # 89567), pMDG.2 and psPAX2 packaging plasmids were used. Lentiviral particles were generated in HEK293FT cells using Lipofectamine 2000 (Thermo Fisher) according to the manufacturer's instructions. 22Rv1 cells were infected with lentiviral supernatant for 24 h and selected with 10 μg/ml of blasticidin (ST018, Beyotime Biotechnology) for 10-14 days. sgRNA sequences targeting −10Kb CTCF site or CTCF sites near *CDK5R2* were designed using CRISPOR (http://crispor.tefor.net), and cloned into the lentiGuide-Puro plasmid (Addgene plasmid # 52963). The sequences of the sgRNAs are shown in Supplementary Data. Lentiviral particles for each sgRNA were generated in HEK293FT cells, and transduced dCas9-KRAB 22Rv1 cells were selected with puromycin for 72 h. To assess the effect of CRISPRi, the relative enrichment of CTCF binding sites was quantified by ChIP-qPCR. *MYC* and *CDK5R2* expression levels were examined by qPCR using primers in Supplementary Information.

## Co-immunoprecipitation assay

Cells were scraped and rinsed twice with PBS. Co-immunoprecipitation assays were performed using the BeaverBeads Protein A/G Immunoprecipitation Kit (22202-100, Beaver Biotechnology) following the manufacturer's protocol. In brief, cells were lysed in IP binding buffer containing protease inhibitors for 30 min and then centrifuged at 12,000 g for 10 min at 4 °C. Protein A/G beads were washed twice and resuspended in IP binding buffer (5 mM Tris-HCl (pH7.4), 150 mM NaCl, 1 mM EDTA, 1% TritonX-100, 5% Glycerol), incubated with anti-Flag, anti-CTCF or anti-MYC antibodies for 30 min at room temperature, and rabbit mAb IgG was used as isotype control. The cell lysates were incubated with the anti-CTCF, anti-MYC, anti-Flag beads, or IgG beads overnight at 4 °C. The antigen-antibody complex was captured by magnetic separation rack and then washed 10 times with washing buffer. Proteins were eluted

from the beads in 50 μL 1×SDS-PAGE loading Buffer, boiled for 5 min, and then subjected to SDS-PAGE and visualized by western blotting using horseradish peroxidase-conjugated mouse anti-rabbit IgG. Monoclonal anti-Myc (IP, 2 μg; Western blot, 1:1000; ab32072; Rabbit monoclonal [Y69]; Lot: GR3377350-5), anti-GFP (IP, 2 μg; Western blot, 1:1000; ab290; Rabbit polyclonal; Lot: GR3321575-1) and anti-MAX (Western blot, 1:1000; ab199489; Rabbit monoclonal [EPR19352], Lot: GR3441065-2) were obtained from Abcam (Cambridge, MA, USA). Antibodies against CTCF (IP, 2 μg; Western blot, 1:1000; 3418 S; Rabbit monoclonal [D31H2]; Lot: 5), Flag (Western blot, 1:1000; 14793; Rabbit monoclonal [D6W5B], Lot: 7), rabbit mAb IgG control (IP, 2 μg; 3900 S; Rabbit monoclonal [DA1E]; Lot: 45), and mouse anti-rabbit IgG mAb (HRP Conjugate) (light-chain specific, 1:1000; 93702 S; Mouse monoclonal [D4W3E]; Lot: 5) were purchased from Cell Signaling Technology (Danvers, MA, USA).

## GST-pulldown

Primers were designed to clone MYC gene into the vector pGEX4T-1 with a GST tag at the N terminus. Escherichia coli BL21(DE3) were transformed with either a plasmid GST-MYC or a pGEX4T-1 empty vector. GST fusion proteins was performed by inducing 100 mL of transformed bacterial cultures with 0.25 mM isopropyl 1-β-d-thiogalactopyranoside and incubating them for 18 h at 16 °C in a shaking incubator. BL21 E. coli were resuspended in ice-cold phosphate-buffered saline with 1 mM PMSF and homogenized by gentle sonication on ice. The soluble proteins were obtained by centrifugation at 10,000 g for 20 min at 4 °C, and the soluble GST or GST-MYC protein were immobilized on glutathione beads (70601-5, Beaver Biotechnology). The Flag-CTCF protein expressing in 22Rv1 cells was then incubated with glutathione beads for 4 h at 4 °C. The beads were again washed five times with cold PBS, and the bead-bound protein complexes were treated with 1× SDS-PAGE loading buffer and detected by Western blotting using an anti-Flag or anti-MAX pAb.

## Proximity ligation assay

Proximity ligation assay (PLA) was carried out to detect a potential physical interaction between MYC and CTCF. To perform this assay, a Duolink® In Situ Orange Starter Kit (Sigma, Cat. No. DUO92102) was used in accordance with the manufacturer's instructions. Briefly, 22Rv1 cells were grown on coverslips in 48-well plates until reaching 60-80% confluence. Then, cells were fixed with 4% paraformaldehyde for 20 min at room temperature and permeabilized with ice-cold 100% methanol for 30 min at −20 °C. Cells were incubated with Duolink® block solution for 60 min at room temperature, and then stained overnight at 4 °C with primary antibodies of different species. In the assay, the antibodies used were mouse anti-MYC (1:50; Santa Cruz, sc-40; Mouse monoclonal [9E10]; Lot: K1920) and rabbit anti-CTCF (1:400; Cell Signaling Technology, 3418 S; Rabbit monoclonal [D31H2]; Lot: 5). The next day, cells were incubated with PLA probes mix containing PLUS antibody and MINUS antibody at 37 °C for 60 min. Then, ligation solution and amplification solution were successively added, respectively for 30 min and 100 min at 37 °C. At last, coverslips were mounted with Prolong Gold mounting medium with DAPI and the images were captured by a confocal microscope (63 × oil immersion; Zeiss LSM 800, Zeiss, Germany). Red spots represented protein-protein interactions.

## Chromosome conformation capture (3C) assay

The 3 C assay was performed using methods as previously described[36]. In brief, 5 million VCaP cells were treated with vehicle or DHT (10 nM) for 24 h and then fixed by 1% formaldehyde in PBS buffer for 10 min, followed by quenching the reaction by glycine. Cells were washed with cold PBS buffer supplemented with 10% FBS, and resuspended in lysis buffer (10 mM Tris-HCl, pH 8.0; 10 mM NaCl; 0.2% NP-40; 1x protease inhibitor). Nuclear extracts were then digested overnight at 37 °C with

400 U PstI (R0140S, New England BioLabs). Digested chromatin DNA was ligated using T4 DNA ligase buffer (750 ul of 10% Triton X-100, 750 ul of 10× NEB ligation buffer, 75 ul of 10 mg/ml BSA, and 4000 U T4 DNA ligase) at room temperature. Proteinase K (20 ul, 19133, Qiagen) was then added and incubated overnight at 65 °C to reverse cross-linking. DNA fragments were purified by ethanol precipitation, and subjected to PCR amplification using the primers listed in Supplementary Data. The ligation products from PstI-digested DNA fragments were used to assess primer efficiency and normalize 3 °C interaction frequency.

## Chromatin immunoprecipitation (ChIP) and ChIP-Seq

ChIP assays were performed using 22RV1 cells with or without MYC stable overexpression. Protein A/G Dynabeads (88845/88847, ThermoFisher) were mixed at a 1:1 ratio, and appropriate antibodies were added to incubated with gentle rotation for 3 hours at 4 °C before immunoprecipitation. Cells were cross-linked by 1% warm formaldehyde for 10 min and then quenched with 125 mM glycine at room temperature. The cell pellets were washed twice with cold PBS, and then samples were incubated on ice with 10 ml of LB1 buffer (50 mM Hepes-KOH, pH 7.5; 140 mM NaCl; 1 mM EDTA; 10% Glycerol; 0.5% NP-40 or Igepal CA-630; 0.25% Triton X-100) for 10 min to extract nuclear fractions. Nuclear fractions were collected by centrifugation and subsequently resuspended in 10 ml of LB2 buffer (10 mM Tris-HCL, pH8.0; 200 mM NaCl; 1 mM EDTA; 0.5 mM EGTA) for 5 min. Nuclear fractions were collected again and resuspended in LB3 buffer (10 mM Tris-HCl, pH 8; 100 mM NaCl; 1 mM EDTA; 0.5 mM EGTA; 0.1% Na-Deoxycholate; 0.5% N-lauroylsarcosine; Protease inhibitor cocktail). Nuclear fractions were transferred to sonication tubes, and sonicated in a water bath sonicator (Diagenode bioruptor) to generate chromatin fragments from 300 bp to 700 bp. 0.1 volume of 10% Triton X-100 was added to each sample. After centrifugation, the supernatant was collected and 10% of the supernatant was used as the input DNA. The rest chromatin lysate was incubated with appropriate antibody-conjugated beads at 4 °C overnight. Antibodies used for ChIP assays are anti-MYC (5 µg, ab32072, Abcam, Rabbit monoclonal [Y69], Lot: GR3377350-5), anti-CTCF (5 µg, 3418 S, CST, Rabbit monoclonal [D31H2], Lot: 5) and anti-H3K27ac (5 µg, ab4729, Abcam, Rabbit polyclonal, Lot: GR3374555-1). Following incubation, beads were washed 10 times with 1 ml of RIPA buffer (50 mM Tris, pH 7.6; 1 M NaCl; 1 mM EDTA; 0.1% SDS; 1% Igepal CA-630; 0.5% sodium deoxycholate), and then resuspended in elution buffer (0.1 M NaHCO3; 1% SDS; proteinase K) to reverse cross-linking of DNA-protein complexes at 65 °C for 8–16 h. DNA was purified using the ChIP DNA Clean & Concentrator kit (D5205, Zymo Reasearch), and then subjected to Illumina ChIP-Seq library construction using ThruPLEX DNA-seq kit (Rubicon Genomics).

## HiChIP

HiChIP was performed as previously described with a few modifications[13]. Ten million cells were collected, pelleted, and resuspended in 1% formaldehyde for 10 min with rotation at room temperature. The crosslinking was then quenched in 125 mM glycine for 5 min. The cells were washed twice by PBS and then resuspended in 500 µL ice-cold Hi-C lysis buffer (10 mM Tris-HCl pH 7.5, 10 mM NaCl, 0.2% NP-40, 1× protease inhibitor) followed by 30 min rotation at 4 °C. Nuclei were pelleted by 2500 rcf centrifugation for 5 min at 4 °C followed by washing once in Hi-C lysis buffer. The nucleus pellets were resuspended in 100 µL 5% SDS and incubated at 62 °C for 10 minutes. To quench the SDS, 335 µL 1.5% Triton X-100 was then added and the mixture was incubated at 37 °C for 15 min. The chromatin was digested by adding 50 µL NEB buffer 2 and 375 U MboI restriction enzyme (NEB, R0147) and incubated at 37 °C for 2 h. The MboI restriction enzyme was then inactivated by 62 °C incubation for 10 minutes. The fill-in master mix (1.5 µL of 10 mM dTTP, 1.5 µL of 10 mM dCTP, 1.5 µL of 10 mM dGTP, 37.5 µL of 0.4 mM biotin-dATP, 10 µL of 5U/µL DNA

Polymerase I (NEB, M0210) was added and the tube was incubated at 37 °C for 1 h with rotation. The ligation mix, containing 10 µL 400 U/µL T4 DNA Ligase (NEB, M0202), 150 µL 10X NEB T4 DNA ligase buffer with 10 mM ATP (NEB, B0202), 125 µL 10% Triton X-100, 3 µL 50 mg/mL BSA, and 660 µL H2O, was added and the tube was incubated at room temperature for 4 h with rotation. The nuclei were pelleted and then resuspended in an 880 µL nuclear lysis buffer (50 mM Tris-HCl pH 7.5, 10 mM EDTA, 1% SDS, 1× protease inhibitor). Nuclei were sonicated in a water bath sonicator (Diagenode bioruptor) to generate chromatin fragments of 300-800 bp. After sonication, the lysate was diluted to 1:9 by adding ChIP dilution buffer (0.01% SDS, 1.10% Triton X-100, 1.2 mM EDTA, 16.7 mM Tris-HCl pH 7.5, 167 mM NaCl). Chromatin immunoprecipitation was performed by adding protein A beads and appropriate antibodies (anti-H3K27ac, 5 µg, ab4729, Rabbit polyclonal, Lot: GR3374555-1; anti-AR, 5 µg, ab108341, Rabbit monoclonal [ER179(2)], Lot: GR3233427-1; anti-CTCF, 5 µg, 3418 S, Rabbit monoclonal [D31H2], Lot: 5) to the sheared chromatin before incubation at 4 °C for overnight with rotation. The ChIPed DNA was eluted from beads by 200 µL ChIP elution buffer (50 mM NaHCO3, 1% SDS) and purified by Zymo DNA Clean & Concentrator kit. The biotin-labelled DNA was captured by streptavidin C1 beads (Invitrogen, 65001) and subjected to tagmentation using 2.5 µL Tn5 (Illumina) for 50 ng DNA. The HiChIP DNA was amplified by ~8 cycles using primers containing Illumina sequencing adapters. Each HiChIP library was sequenced by an Illumina sequencer to the depth of ~150 million 2 × 150 bp paired-end reads.

## HiChIP data processing

Pair-end HiChIP reads were mapped to hg19 human reference genome by Bowtie2[37] within the HiCUP pipeline (v0.7.2)[38]. The digested reference genome was created by HiCUP using the MboI cutting site. The resulting *bam* files from HiCUP pipeline were converted into valid fragment pairs by samtools[39] and bedtools[40]. The HiChIP valid pairs and matched ChIP-Seq peaks were then subjected to the hichipper pipeline (v0.7.7)[41] to generate *bedpe* files containing HiChIP loops. For HiChIP loops called by hichipper, the median loop anchor width is ~2.5 Kb for libraries prepared using MboI enzyme. In this study, only the filtered intra-chromosome loops were used for downstream analyses.

## Significant loop calling

To identify high confidence HiChIP loops, we considered that the mated PE reads could be located at any possible loop anchor pair by a binomial distribution. The number of all possible loops ($N_{Possible\ loops}$) are a combination of two anchors from all possible loop anchors ($N_{Possible\ loop\ anchors}$) within the same chromosome. The $N_{Possible\ loop\ anchors}$ can be calculated by:

$$N_{Possible\ loop\ anchors} = N_{Observed\ loop\ anchors} + N_{Potential\ loop\ anchors} \quad (1)$$

The observed loop anchors are loop anchors with mated PE reads, which could be found in the output of the analysis pipeline. Potential loop anchors are genomic regions with matched ChIP peaks but no mated PE reads. Since the HiChIP loop anchor could span multiple ChIP-Seq peaks, we cannot use the number of ChIP-Seq peaks as the amount of potential loop anchors. Intuitively,

$$\frac{N_{Observed\ loop\ anchors}}{N_{Possible\ loop\ anchors}} \propto \frac{N_{ChIP\ peaks\ in\ observed\ loop\ anchors}}{N_{Total\ ChIP\ peaks}} \quad (2)$$

Then we can roughly estimate the number of overall loop anchors (possible loop anchors) by:

$$N_{Possible\ loop\ anchors} \approx \frac{N_{Observed\ loop\ anchors} \times N_{Total\ ChIP\ peaks}}{N_{ChIP\ peaks\ in\ observed\ loop\ anchors}} \quad (3)$$

Because we only estimate the intra-chromosome loops, we have to calculate $N_{Possible\,loop\,anchors}$ for each chromosome and get the sum. Therefore, the $N_{Possible\,loops}$ can be presented by:

$$N_{Possible\,loops} = \sum_i C\left(N^i_{Possible\,loop\,anchors}, 2\right)$$

$$= \sum_i C\left(\left[\frac{N^i_{Observed\,loop\,anchors} \times N^i_{Total\,ChIP\,peaks}}{N^i_{ChIP\,peaks\,in\,observed\,loop\,anchors}}\right], 2\right) \quad (4)$$

$$i = chr1, chr2, \cdots, chrX$$

According to the binomial distribution, the possibility of a given loop can be calculated as

$$P(X = m) = C(n, m)p^m(1 - p)^{n-m} \quad (5)$$

$m$ denotes the loop strength corrected by anchor length. $n$ denotes the total corrected loop strength. $p$ is the probability of one PE mated read mapped to a specific loop. $p$ denotes the determined by:

$$p = \frac{1}{N_{Possible\,loops}} \quad (6)$$

$P$-values are corrected by Bonferroni-Holm method and loops with adjusted $P$-values < 0.05 were considered as reliable loops in this study.

## Visualization
HiChIP loops at a focal genomic region were visualized using R. Normalized loop strength was represented by loop curve length. For each loop, the centers of two anchors were used as the start and end of a loop curve. To visualize chromosome-wide HiChIP contact map, HiChIP *fastq* files were processed by Juicer (v1.6) pipeline[42] to generate *hic* files. The *hic* files were loaded into R by plotgardener[43] to show the genomic contact heatmap.

## Motif analysis
To identify DNA binding motifs enriched in the open chromatin regions of HiChIP loop anchors, TCGA PCa ATAC-Seq peak set was downloaded from https://gdc.cancer.gov/about-data/publications/ATACseq-AWG[44]. The hg38 version PCa ATAC-Seq peaks were converted to hg19 genome coordinates using "hg38ToHg19.over.chain" from UCSC genome browser. The motifs enriched in the indicated regions were then identified by Cistrome SeqPos[45]. For the motif analysis in H3K27ac and/or CTCF anchors with or without MYC binding, 'findMotifsGenome.pl' script of HOMER (v4.11)[46] was used to obtain the motifs enriched in six peak types with parameters '-size 600 -mask' and 'annotatePeaks.pl' script was used to calculate the distribution of MYC or CTCF DNA binding motif PWMs at indicated peaks.

## Loop anchor annotation and enrichment analysis
The genomic features and nearby genes were annotated to HiChIP loop anchors by the annotatePeak function of R packages ChIPseeker[47] and TxDb.Hsapiens.UCSC.hg19.knownGene. To identify the pathways related with loop anchors, the anchor-associated genes were subjected to KEGG pathway enrichment analysis using R package clusterProfiler[48].

## Hi-C data processing
22RV1 Hi-C data was downloaded from GSE118629[49]. By the Juicer (v1.6) pipeline[42], the raw Hi-C read pairs were first mapped to hg19 reference genome, and after deduplication, the *bam* files of alignments were used to generate *hic* files, which contain genomic interaction matrices. The normalized average interaction strength of aggregated loops was

then obtained from *hic* files by Aggregate Peak Analysis (APA) function of Juicer.

## ChIP-Seq data processing
Reads from ChIP-Seq experiments were aligned to the hg19 version of the human reference genome by Bowtie2 (version 2.2.1). The resulting *sam* files were converted to *bam* files by samtools (v.0.1.18). MACS2 (v2.2.7.1)[50] was used to call peaks from *bam* files with parameters '–keep-dup=1 -g hs -B–SPMR'. The resultant bedGraph files containing signal per million reads were converted to bigWig files by UCSC tools (v385). The bigWig files were loaded to genome browser IGV (v2.8.12) for peak binding visualization. Deeptools (v3.4.3)[51] was used to extract ChIP-Seq signals of indicated peaks from bigWig files and generate the profile plots.

## RNA-Seq data processing
The reads were aligned to the hg19 human reference genome by STAR (version 2.4.2a) with default settings[52]. The resulting *.ReadsPerGene.out.tab* files were then merged to a read count matrix and the matrix was used for differential expression analysis by R package DESeq2. GO enrichment analysis and GSEA were performed by R package clusterProfiler[48]. The gene expression levels were also quantified by calculating the reads per kilobase per million mapped reads (RPKM) using the read count matrix and GENCODE v19 gene annotation.

## Reporting summary
Further information on research design is available in the Nature Portfolio Reporting Summary linked to this article.

## Data availability
Hg19 human reference genome was downloaded from NCBI [https://www.ncbi.nlm.nih.gov/assembly/GCF_000001405.13/].The following publicly available ChIP-Seq and ATAC-Seq data sets used in this paper were obtain from GEO: LNCaP H3K27ac ChIP-Seq (GSM1249448)[53], 22Rv1 H3K27ac ChIP-Seq (GSM2827407)[18], 22Rv1 CTCF ChIP-Seq (GSM2828839)[18], VCaP AR ChIP-Seq (GSE55062)[54], VCaP CTCF ChIP-Seq (GSE84432)[55], GM12878 CTCF ChIP-Seq (GSM935611)[18], Hela CTCF ChIP-Seq (GSM2915166)[16], A549 RAD21 ChIP-Seq (GSM3106369)[18] and 22Rv1 ATAC-Seq (GSM3075372)[56]. To determine the relative peak binding affinities among cell lines, the CTCF ChIP-Seq data of 22Rv1 [https://www.encodeproject.org/experiments/ENCSR857PBV/], C4-2B [https://www.encodeproject.org/experiments/ENCSR460LGH/], LNCaP [https://www.encodeproject.org/experiments/ENCSR315NAC/] and VCaP [https://www.encodeproject.org/experiments/ENCSR265ARE/] cells and H3K27ac ChIP-Seq data of 22Rv1 and VCaP cells were also downloaded from ENCODE (https://www.encodeproject.org). The RNA-Seq data of 22Rv1, C4-2B, LNCaP and VCaP cells were obtained from CCLE (https://portals.broadinstitute.org/ccle/home) and GEO (GSE25183)[57]. The combined ATAC-Seq peaks (hg38) of TCGA-PRAD samples were downloaded from GDC[44] and lifted over to hg19 by UCSC 'hg38ToHg19.over.chain'. RNA-Seq and DNA methylation data of Changhai 2020 PCa cohort[58] were obtained from www.cpgea.com. GM12878 CTCF and Hela CTCF HiChIP data were obtained from GEO by accession numbers GSM3424974[17] and GSM2974085[16], respectively. The RNA-Seq, H3K27ac ChIP-Seq and MYC ChIP-Seq data of VCaP cells we generated in another study were deposited to GEO under GSE157104[14]. All the RNA-Seq, ChIP-Seq and HiChIP data we generated for this study were deposited to GEO under GSE172498 and GSE200168. Source data are provided with this paper. The remaining data are available within the Article, Supplementary Information or Source Data file. Source data are provided with this paper.

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

## Acknowledgements

This work was supported by National Key Research and Development Project of China (2019YFA0111400 to H.G.), Taishan Scholar Program of Shandong Province (NO.tsqn201812136 to H.G.), the Natural Science Foundation of Shandong Province, China (ZR2021YQ49 to H.G. and ZR2019PH021 to Z.W.), the National Natural Science Foundation of China (82173052 to H.G. and 81901421 to Z.W.), NSERC discovery grant (498706 to H.H.H.), CIHR operating grants (142246, 152863, 152864 and 159567 to H.H.H.), Terry Fox New Frontiers Program Project Grant (1090 P3 to H.H.H.), Funding for Study Abroad Program by the Government of Shandong Province (NO.201803012 to Z.W.). H.H.H. holds Joey and Toby Tanenbaum Brazilian Ball Chair in Prostate Cancer Research.

## Author contributions

Z.W., H.H.H., and H.G. designed the studies and wrote the manuscript; Z.W. and H.G. performed the experiments with help from S.W., Y.X., W.W., F.S., E.O., X.X., X.L., Z.L., L.D., Y.W., S.Y.C., and C.W.; Z.W., H.H.H. and H.G. conducted the data analysis with help from M.A., P.S., T.W., T.J., Y.Z. and S.J.C.; H.G., H.H.H., Z.W., S.Y.C., X.X., Y.Z., and F.S. revised the manuscript.

## Competing interests

The authors declare no competing interests.
