## [Peer Review File · Nature Communications]

MYC reshapes CTCF-mediated chromatin architecture in prostate cancerReviewers' Comments:

Reviewer #1:

Remarks to the Author:

The functional link between oncogenic transcription factor (TF) Myc and chromatin architecture organizer CTCF has not been established in prostate cancer (PCa). In this study, Wei et al. first characterized regulatory elements (RE)-associated chromatin interactions and androgen-induced chromatin dynamics in multiple PCa cells by integrative analysis of AR, H3K27Ac HiChIP, matched ChIP-seq and RNA-seq data, which revealed the chromatin landscapes involved in androgen-regulated gene expression in PCa. Further motif analysis of AR+/H3K27ac+ and H3K27ac-only loop anchors underlined the fundamental roles of CTCF in establishing RE interaction in PCa. HiChIP analysis showed that CTCF-mediated chromatin interactions were cell-type specific and responsible for PCa-related gene expression. H3K27ac- CTCF sites were associated with suppression of their looped essential genes, which further led to the discovery of a CTCF region located at ~10 Kb upstream of the MYC gene. Deletion of this CTCF regions induced not only local but also genome-wide alterations of chromatin architecture involving CTCF and H3K27ac loops. Interestingly, MYC interacted with CTCF and facilitated its chromatin occupancy at CTCF/MYC common sites. MYC overexpression significantly led to gene expression repression by synergistically strengthening CTCF binding/loops and impairing H3K27ac-associated interactions. Overall, this is a very comprehensive study to elucidate novel functions of CTCF and MYC in 3D genome organization of prostate cancer. A couple of comments below need to be addressed to improve the manuscript.

1. The authors provided detailed genomic evidence that deletion of a CTCF site upstream of MYC gene disrupted CTCF loop and induced extensive new H3K27ac-associated loops, the function of this CTCF site should be determined to link its genomic functions with cell phenotypes. The gel images should be provided to demonstrate the successful deletion of this site.
2. In line 286-287, how the authors defined super-enhancer within PCAT1/2 region?

Reviewer #2:

Remarks to the Author:

In this work Zhao Wei and collaborators investigated the epigenome the role of MYC in tuning the 3D chromatin organization in prostate cancer cell lines. They focus on determining whether MYC could affect the chromatin looping between cis regulatory elements through its interaction with the architectural protein CTCF, which in principle could affect its binding affinity to chromatin. Although the proposed mechanism is of high relevance in the field of chromatin biology, the conclusions are not appropriately supported by the presented data. Indeed, most of the conclusions are drawn on correlative analyses of genome wide approaches, whose interpretations can be put into discussion (see below). Indeed, there are important technical limitations that reduce the impact and the robustness of this study.

Major criticisms:

1. In characterizing chromatin loop associated with H3K27ac and AR enrichment in VCAP cells upon treatment with DHT,(Fig.1) the author do not clarify what is the resolution (bin size) adopted for these analyses, nor the fraction of shared regions (bins) between H3K27ac and AR, retrieved from ChIP-seq or HiChIP. These information are necessary to be able to interpret the presented results.
2. Related to figure 1C, what is the fraction of H3K27ac and AR-associated loops not anchored to "expressed genes"? How to they identified "expressed genes", in terms of bins? How do they manage bins which contains more than one anchors and or more than one CREs?
3. The conclusion that" AR-associated chromatin contacts are rapidly formed upon androgen stimulation" is not supported by the presented data. This is an overinterpretation of the measured changes in chromatin looping frequency (named by the author as strength) observed upon treatment with DHT, at AR-enriched loop size. They should measure the number of new loops that actually would be detected in response of DHT treatment and normalize it on the relative changes of AR chromatin

binding. Does AR binding affinity changes in response to DHT treatment? If so (and this would be expected), how can they distinguish between an effective increment of chromatin looping with respect an increase IP of the chromatin-bound TF?

4. The correlation between increase of frequency of AR chromatin looping with expressed (induced?) genes upon DHT treatment should be confirmed using an independent and orthogonal approach. For example for the two mentioned genes, they should perform either 4C assay and/or FISH-based experiments.

5. The change in called loops could results as a consequence of a) abundance of the IP factor or b) its chromatin affinity; c) overall increase of the looping in all cells or d) formation of looping in a fraction of cells. How do the author distinguish between these possible explanations for the observed changes in chromatin looping?

6. In fig.2 the author presented CTCF HiCHIP data through which the recall a very low number of loops (with respect to what have been obtained in other work either by Hi-C or HiCHIP): how do they explain this? Plus the modest overly of common loops may results by the intrinsic variability (and low reproducibility) of the technique: what is the reproducibility of the results in each cell type? The results should be shown not as merged data but comparing the reproducibility of the technique by comparing the replicate of each cell line. Only After ensuring that the data are reproducible the author can perform any comparison between different cell lines.

7. Based on the above comment, the differential analyses presented in Fig. 3a make no sense. In addition, as the are no 3 biological replicates the robustness of these results is put into discussion.

8. It is surprising observing that the CTCF HiCHIP detected so few (upstream) loops in the surrounding of MYC gene: did the author compared by any means they results with the available published results? Similar (prostate) cancer cells show much broader and frequent chromatin looping. Can the author explain why such a large difference in their dataset?

9. The author claimed that the deletion of the proximal (-10kb) CRE perturbs MYC abundance and its relative chromatin affinity (Fig 3 c,d). They do not provide sufficient data supporting this conclusion: could they demonstrate by WB that MYC protein abundance is increased? Could they provide data showing MYC chromatin binding is altered? How many binding sites do they find? What is the level and the distribution of MYC binding genome-wide? Which genomic region showed an increment of MYC binding? How did they determine MYC occupancy?

10. In fig. 3e,f the authors used TF binding motif analyses to conclude that "high MYC expression facilitate CTCF-CTCF loop formation". They rely on the different enrichment of MYC motif in the proximity of CTCF motifs within the CTCF-only anchors, with respect to CTCF and H3K27ac anchors. This kind of differential enrichment for a given motif can suffer a lot based on the composition and complexity of the compared genomic regions. Are the number of anchors comparable in these two samples? Is the level of CTCF binding comparable? If so how do they justify the result that indicate that CTCF motifs is differentially enriched in this comparison?

11. The comparative analyses performed upon OE of MYC, brought to the general conclusion that MYC facilitate CTCF chromatin occupancy at the shared sites. Once again this is an over-interpretation of the obtained results. As shown in panel 4f, there is already a clear difference in the relative abundance of CTCF binding, depending which genomic regions have been considered, indicating that CTCF occupancy is different among the anchors, depending on the chromatin context (H3K27ac, in this case). The measured higher abundance of CTCF signal (from ChIP-seq) could depend by multiple regions, apart an increase in chromatin occupancy, which relates to an increase of the affinity and/or a change in the resident time. As all these results relies on population-based methodology, it may simply reflect that more cells at that given time of the analyses showed occupancy of CTCF. Plus, MYC alters multiple biological functions, including the cell cycle. Given that genome organization (including TADs) reflects replication timing, it is more that reasonable that MYC OE perturb the genome replication pattern (or time of division), thereby explaining the modes increment of CTCF binding.

12. The Co-IP experiments (Fig 4h) lack important control to support the conclusion that MYC and CTCF are interacting. For example, do the author consider the possibility that chromatin bridging affected their results? The experimental conditions (buffer composition for example) are not clear, thereby is difficult to determine which ionic strength have been used in this assay. They need to include positive and negative controls to determine the specificity of this assay, including known MYC

interactors (apart MAX). How do they explain that the OE of MYC did not change the level of immunoprecipitated MYC and CTCF? If they imply cooperativity between these two TFs, which domain/region of MYC and CTCF participate in this interaction? Finally orthogonal assays as to be included to support the conclusion that MYC and CTCF can interact within cells (like PLA or direct measurement of interaction like FRET experiments).

13. Though correlative analyses, the author concluded that MYC suppresses neuroendocrine gene expression by enhancing CTCF chromatin looping (Fig 5). However they need to provide supportive data to this conclusion. For example, they should perturb CTCF (or MYC) binding to one of this anchor site by point mutation to then determine the relative change in gene expression. Is indeed expected from the claimed conclusion that the cooperation between MYC and CTCF on these specific genomic regions being causative of the increased chromatin looping and thereby the diminished gene expression. If so, the author should demonstrate this directly by perturbing the binding of these TFs on this site to then determine the relative change of chromatin looping and gene expression. This is the main point of their work, thereby they have to provide supportive (and not correlative) data.

Reviewer #3:

Remarks to the Author:

The main and most important finding in this study is that Myc, by directly interacting with CTCF, can alter the CTCF-mediated 3D genome organization in prostate cancer cell lines. The authors utilized multiple, publicly available genomics datasets, but also performed a great amount of HiChIP experiments and bioinformatic analyses.

Several pieces of information and a great number of conclusions that are presented here as novel have been previously published, either by the authors or other groups (MYC overexpression leads to increased chromatin interactions at superenhancers and MYC binding sites, doi:10.1101/gr.276313.121, Genome Res. 2022, or Transcriptional Dysregulation of MYC Reveals Common Enhancer-Docking Mechanism, <https://doi.org/10.1016/j.celrep.2018.03.056>). Therefore, it would be important to properly present what is already known in the field and nicely uncover the novel information of this manuscript.

The following points would require further clarification in the manuscript:

1. Line 60: What CTCF interactome the authors refer to? Is it the protein interactome or the CTCF-dependent 3D genome organization of PCA cells?
2. For example a great number of findings, presented in the manuscript, have already been previously published:
 - CTCF binding site located 2 kb upstream of the MYC promoter, <https://www.ncbi.nlm.nih.gov/entrez/eutils/elink.fcgi?dbfrom=pubmed&retmode=ref&cmd=prlinks&id=29641996>, <https://doi.org/10.1016/j.celrep.2018.03.056>
 - Nucleic Acids Research, Volume 47, Issue 13, 26 July 2019, Pages 6699–6713, <https://doi.org/10.1093/nar/gkz462>
3. More information would be required in the introduction regarding the use of DHT, or androgen induction. This would be helpful for following the results section.
4. Fig.1h: the AR-ChIPseq, hours of DHT treatment is not indicated (2h or 24h?).
5. Fig.S1j,k: The fact that H3K27ac loops are changed or not would be better supported by quantitation instead of only visual examination. Quantitative data on loop-strength for these gene loci would be appreciated.
6. Lines 166-170: please elaborate on the working hypothesis regarding the androgen-induced redistribution of cofactors. For serving the flow of the text in the manuscript, it is not clear why androgen-induced transcription repression is examined.
7. Line 197: it is not described what the cell lines under study are, what is their origin, in order to understand the differences in HiChIP experiments. For example, why is there so much difference in the CTCF HiChIP results for VCaP and 22Rv1 cell lines?
8. Line 200: Figure 2a is called which is displaying ChIPseq results for CTCF and not HiChIP anchors.

Please correct.

9. Fig.2a: Please check graphs, they look almost the same although they depict CTCF loops in different cell lines.

10. Fig.S2e: It is not clear what is depicted. Although, not H3K27ac ChIPseq was used but rather H3K27ac HiChIP experiment, is the overlap between K27ac anchors with CTCF anchors indicated? This is not clear in every case (Supplementary Fig. 1,2). Please clearly indicate what each figure depicts. Are they H3K27ac HiChIP anchors, or H3K27ac ChIPseq peaks? The same for all transcription factors used.

11. Fig. S2f: The figure legend does not provide any information on how the promoter is defined so that distance categories are assigned. Based on this graph, over 80% of CTCF loop anchors are localized on genes and their promoters.

12. Lines 223-227, Fig. 2d: the conclusion drawn is an overestimation and the data presented do not support it. Please elaborate.

13. Fig.2e: the conclusion is not clear (lines 230-236). There is no cell-type specific enrichment of cancer-associated gene pathways for genes localized to CTCF HiChIP anchors (the criteria for proximity are not described).

14. Lines 286-287: The choice of the PCAT1/2 gene and the reference to its super-enhancers is not justified.

15. Fig. S3c: Myc locus is localized on chromosome 8, though upon deletion of the -10Kb element, bigger effects are observed in CTCF looping in other chromosomes compared to the anticipated cis-effects. Taken under consideration the Myc hypothesis, one would expect in Fig.3c to present an analysis of the CTCF loops presented in SFig.3c. Moreover, in Fig.3d is there really an obvious difference in Myc binding in CTCF anchors in the samples +/- the -10Kb deletion?

16. HiChIP protocol: Restriction enzyme digestion takes place for only 20 minutes? Immunoprecipitation takes place in 1% SDS (Nuclear Lysis Buffer)?

Response to reviewer's comments:

We thank the reviewers for their critical reading of our manuscript and for their suggestions on how to further improve it. Below are our detailed responses (in blue) to each of the reviewer's specific comments.

Reviewers' comments:

Reviewer #1 (Remarks to the Author):

The functional link between oncogenic transcription factor (TF) Myc and chromatin architecture organizer CTCF has not been established in prostate cancer (PCa). In this study, Wei et al. first characterized regulatory elements (RE)-associated chromatin interactions and androgen-induced chromatin dynamics in multiple PCa cells by integrative analysis of AR, H3K27Ac HiChIP, matched ChIP-seq and RNA-seq data, which revealed the chromatin landscapes involved in androgen-regulated gene expression in PCa. Further motif analysis of AR+/H3K27ac+ and H3K27ac-only loop anchors underlined the fundamental roles of CTCF in establishing RE interaction in PCa. HiChIP analysis showed that CTCF-mediated chromatin interactions were cell-type specific and responsible for PCa-related gene expression. H3K27ac- CTCF sites were associated with suppression of their looped essential genes, which further led to the discovery of a CTCF region located at ~10 Kb upstream of the MYC gene. Deletion of this CTCF regions induced not only local but also genome-wide alterations of chromatin architecture involving CTCF and H3K27ac loops. Interestingly, MYC interacted with CTCF and facilitated its chromatin occupancy at CTCF/MYC common sites. MYC overexpression significantly led to gene expression repression by synergistically strengthening CTCF binding/loops and impairing H3K27ac-associated interactions. Overall, this is a very comprehensive study to elucidate novel functions of CTCF and MYC in 3D genome organization of prostate cancer. A couple of comments below need to be addressed to improve the manuscript.

We thank the reviewer for the positive evaluation of our work.

1. The authors provided detailed genomic evidence that deletion of a CTCF site upstream of MYC gene disrupted CTCF loop and induced extensive new H3K27ac-associated loops, the function of this CTCF site should be determined to link its genomic functions with cell phenotypes. The gel images should be provided to demonstrate the successful deletion of this site.

We thank the reviewer for this suggestion. We have added the original gel image showing that the “-10Kb CTCF site” was successfully deleted by CRISPR-based paired-sgRNAs in **Figure S4b** (also shown above).

Additionally, we validated the function of “-10Kb CTCF site” by CRISPRi. In **Figure S4f-j** (also shown as above), CTCF occupancy was dramatically reduced by dCas9-KRAB complex guided by a sgRNA targeting this site (**f**). Both MYC mRNA and protein levels were upregulated upon inhibition of “-10Kb CTCF site” (**g**). Consistent with MYC expression elevation, CRISPRi of “-10Kb CTCF site” also increased MYC target gene expression (**j**), MYC binding at target gene promoters (**i**), and cell proliferation (**h**).

2. In line 286-287, how the authors defined super-enhancer within PCAT1/2 region?

In a previous study (Guo et al., Nat Commun, 2021, PMID: 34911936), we identified three super-enhancers within PCAT1/2 region based on H3K27ac ChIP-seq peaks by Ranking Of Super Enhancer (ROSE) and performed 3C-ddPCR to validate that these super-enhancers interact with MYC promoter to regulate MYC expression in VCaP cells. Our findings are consistent with reports that in prostate cancer cells (LNCaP and VCaP) the PCAT1/2 sites physically and functionally interact with MYC promoter (PMID: 20453196; PMID: 31735626). In this study, the distal chromatin interaction was reproduced in 22Rv1 cells. We added a citation of our previous work to introduce these super-enhancers at line 304~306 (as shown below).

“We and others previously reported that a cluster of super-enhancers within PCAT1/2 region interact with MYC promoter to regulate MYC expression in VCaP cells^{14,21,22}. Here, we found the super-enhancers within PCAT1/2 region were also robustly looped to MYC in 22Rv1 cells, but the looping between them was not significantly changed by -10Kb CTCF site deletion (Figure S4d).”

Reviewer #2 (Remarks to the Author):

In this work Zhao Wei and collaborators investigated the epigenome the role of MYC in tuning the 3D chromatin organization in prostate cancer cell lines. They focus on determining whether MYC could affect the chromatin looping between cis regulatory elements through its interaction with the architectural protein CTCF, which in principle could affect its binding affinity to chromatin.

Although the proposed mechanism is of high relevance in the field of chromatin biology, the conclusions are not appropriately supported by the presented data. Indeed, most of the conclusions are drawn on correlative analyses of genome wide approaches, whose interpretations can be put into discussion (see below). Indeed, there are important technical limitations that reduce the impact and the robustness of this study.

We thank the reviewer for the recognizing the impact of our work in the field and thoughtful suggestions on how to improve our manuscript. To further assess the conclusions we made, we have performed a substantial amount of additional experiments and analyses, as included in the updated manuscript and in our responses below.

Major criticisms:

1. In characterizing chromatin loop associated with H3K27ac and AR enrichment in VCAP cells upon treatment with DHT,(Fig.1) the author do not clarify what is the resolution (bin size) adopted for these analyses, nor the

fraction of shared regions (bins) between H3K27ac and AR, retrieved from ChIP-seq or HiChIP. These information are necessary to be able to interpret the presented results.

We thank the reviewer for this suggestion. The resolution definition in HiChIP data analysis is different from Hi-C data analysis. Hi-C analysis pipelines like Fit-HiC, and HiCCUPS bin genome into fixed intervals. For HiChIP analysis using hichipper, the resolution size was the median loop anchor width, which is ~2.5 Kb for most HiChIP data based on the Mbol enzyme. We now added the loop anchor size information to the Method section.

We now retrieved the overlapping information between H3K27ac loop anchors and AR anchors. In **Figure S1d** (as shown above), 37.0% of H3K27ac anchors overlap with 50.1% of AR anchors, which is consistent with our observation that a large fraction of H3K27ac loops is independent of AR.

2. Related to figure 1C, what is the fraction of H3K27ac and AR-associated loops not anchored to “expressed genes”? How to they identified “expressed genes”, in terms of bins? How do they manage bins which contains more than one anchors and or more than one CREs?

We thank the reviewer for these questions.

We first defined the expressed genes as genes of average FPKM > 0.1 in VCaP cell RNA-Seq data. To assign expressed genes anchored by H3K27ac or AR-associated loops, we analyzed the overlapping between gene transcription start sites (TSSs) and loop anchors. If the left anchor of one loop overlaps with a TSS, the normalized strength of this loop was accumulated to the TSS downstream anchor location that spans this loop. Similarly, if the right anchor of the loop overlaps with a TSS, the normalized strength was accumulated to the corresponding upstream anchor location. If none of the anchors of this loop overlaps with this TSS, the strength of this loop was not counted for this TSS.

Since our distal anchor reads distribution analysis is TSS-centric, anchors from the same loop may overlap with more than one TSSs. In that case, we calculated the contribution of this loop strength to different TSSs separately, and sum up the location-labeled distal anchor strength of all TSSs. Because we applied the same strategy for three conditions (Veh, DHT_2hr, DHT_24hr) in **Figure 1c**, the results are justified for comparison among conditions.

We now added the information of loops anchored to expressed gene TSSs in **Figure S1g** (also shown above).

3. The conclusion that "AR-associated chromatin contacts are rapidly formed upon androgen stimulation" is not supported by the presented data. This is an overinterpretation of the measured changes in chromatin looping frequency (named by the author as strength) observed upon treatment with DHT, at AR-enriched loop size. They should measure the number of new loops that actually would be detected in response of DHT treatment and normalize it on the relative changes of AR chromatin binding. Does AR binding affinity changes in response to DHT treatment? If so (and this would be expected), how can they distinguish between an effective increment of chromatin looping with respect an increase IP of the chromatin-bound TF?

Review figure 1

The reviewer raised a great question of whether the TF-associated loop strength changes should be distinguished from TF binding changes at loop anchors in HiChIP analysis. Theoretically, correlation is expected between TF-associated chromatin loop strength and TF binding affinity; however, not every random TF binding site pair form solid chromatin loop, as exemplified in **Figure 4Sa**. Strong TF binding at two anchors is necessary but not sufficient to establish strong chromatin loop of the two anchors. In other words, TF-binding is only one of the important factors in TF-associated loop establishment, and potentially could be more important to determine the loop strength once the loop has been established. Indeed, we analyzed AR-loop strength changes and AR binding changes at loops anchors, and found they are highly correlated; but there are also a few loops independent of TF binding changes (**Review figure 1**, as shown above). We now revised this sentence as “AR-associated chromatin contacts are significantly strengthened upon androgen stimulation”.

4. The correlation between increase of frequency of AR chromatin looping with expressed (induced?) genes upon DHT treatment should be confirmed using an independent and orthogonal approach. For example for the two mentioned genes, they should perform either 4C assay and/or FISH-based experiments.

We thank the reviewer for this suggestion. We have performed 3C-qPCR to validate the chromatin interaction between IL20RA promoter and enhancers. As shown in **Figure 1h** (shown above), all three enhancers have significant interaction with IL20RA promoter compared with genomic background, which is consistent with H3K27ac HiChIP results at Veh condition. These interactions are further reinforced upon DHT treatment, consistent with the DHT-induced AR loops. Since 3C results reflect the overall chromatin interaction, the trend of 3C-qPCR is not exactly the same as H3K27ac or AR HiChIP, but a mixture of the two.

5. The change in called loops could result as a consequence of a) abundance of the IP factor or b) its chromatin affinity; c) overall increase of the looping in all cells or d) formation of looping in a fraction of cells. How do the author distinguish between these possible explanations for the observed changes in chromatin looping?

We thank the reviewer for this thoughtful comment. HiChIP captures the chromatin occupied by the specific IP factor (TF), whose DNA affinity and proximity would contribute to HiChIP signals. Although changes in protein expression of the TF may impact the TF binding affinity, the changes of loops should still be directly linked with the TF binding affinity. Therefore, we would attribute the origin of HiChIP loop changes to the TF binding affinity rather than its protein abundance.

HiChIP experiments in tissues is often arguable that the changes of called loops could reflect the formation of looping in a fraction of cells, with less concerns in cell lines due to relatively higher the homogeneity in treatments and assays. Nevertheless, we agree single-cell Hi-C/HiChIP approach would help to determine whether major loop changes reflect looping formation/disruption in a

larger proportion of cells compared with minor loop changes; however, obviously it is out-of-scope for the current study.

6. In fig.2 the author presented CTCF HiChIP data through which the recall a very low number of loops (with respect to what have been obtained in other work either by Hi-C or HiChIP): how do they explain this? Plus the modest overly of common loops may results by the intrinsic variability (and low reproducibility) of the technique: what is the reproducibility of the results in each cell type? The results should be shown not as merged data but comparing the reproducibility of the technique by comparing the replicate of each cell line. Only After ensuring that the data are reproducible the author can perform any comparison between different cell lines.

We appreciate the reviewer for pointing this out and it is indeed a great question.

In previous version, we used two different CTCF antibodies in HiChIP experiments. Antibody ab70303 was used for CTCF HiChIPs in **Figure 2 and 3**, and antibody 3418S in **Figure 5**. Indeed, compared with 3418S, ab70303 recalls lower numbers of CTCF loops. To address both the antibody efficiency and data reproducibility concerns, we have re-performed the CTCF HiChIP experiments in both VCaP and 22Rv1 cells, using the 3418S antibody with two replicates for each cell line.

As shown in **Figure 2a-c**, the replicates of the same cell line are highly correlated ($r = 0.9318$ for VCaP replicates; $r = 0.9271$ for 22Rv1 replicates), while the samples between different cell lines were less correlated ($r = 0.6958$ and 0.6720), indicating the reliability of these CTCF HiChIP datasets. These datasets have been added to GEO deposit (GSE200168), and the accession numbers are GSM6856446, GSM6856447, GSM6856448 and GSM6856449.

a

CTCF HiChIP loops	
Sample	Significant loops (FDR < 0.05)
VCaP rep-1	127197
VCaP rep-2	114435
22Rv1 rep-1	159462
22Rv1 rep-2	164907
Hela (GSM2974085)	121205
GM12878 (GSM3424974)	193825

b
To further confirm the data quality, we compared these data with published CTCF HiChIP data (Mumbach MR, et al., Nat Methods, 2019, PMID: 31133759). As shown above, our 22Rv1 CTCF HiChIP data using the 3418S antibody showed good correlations with published CTCF HiChIP data ($r = 0.7171$ with GM12878 cells; $r = 0.6738$ with Hela cells; **Figure S3b**). The significant CTCF loop numbers of our data are comparable to those in published data (**Figure S3a**). Together, these results validate the quality of our new CTCF HiChIP data.

We next repeated the analyses in Figure 2 using new CTCF HiChIP data based on 3418S antibody. In revised **Figure 2d-h and related panels in Figure S3** (as shown above), all the biological relevant analyses showed very similar trends in comparison to previous CTCF HiChIP profiling based on the ab70303 antibody. Collectively, these results suggest although our ab70303 CTCF HiChIPs recalled relatively small numbers of CTCF loops, they did capture the biological relevant loops.

7. Based on the above comment, the differential analyses presented in Fig. 3a make no sense. In addition, as there are no 3 biological replicates the robustness of these results is put into discussion.

As described in our response to above comment #6, we have validated the quality and reproducibility of our new CTCF HiChIP datasets. We then repeated the analysis in **Figure 3a** using the new CTCF HiChIP datasets and obtained comparable results to previous one (shown above). In the updated manuscript, we updated **Figure 3a** with new analysis.

8. It is surprising observing that the CTCF HiChIP detected so few (upstream) loops in the surrounding of MYC gene: did the author compared by any means their results with the available published results? Similar (prostate) cancer cells show much broader and frequent chromatin looping. Can the author explain why such a large difference in their dataset?

a

We thank the reviewer for the questions. To highlight the disruption of CTCF looping between -10Kb site and MYC promoter upon -10Kb site CRISPR deletion, we only showed the significant changed CTCF loops in a small highlighted genomic region (~48 Kb) surrounding MYC locus, resulting in very few selected loops. To display a broader landscape of CTCF-mediated looping surrounding MYC region, we now plotted CTCF loops of the 8q24 region based on our new CTCF HiChIPs (3418S antibody), together with two published CTCF HiChIPs. In **Figure S4a** (shown above), our VCaP and 22Rv1 data have similar numbers of CTCF loops in this region compared with datasets of GM12878 and HeLa cells, and there are also reasonable numbers of common loops between our datasets and published datasets. Interestingly, the long range CTCF loops between chr8: 127.83 Mb and chr8: 128.738 Mb were captured in both VCaP and 22Rv1 CTCF HiChIP datasets, but not in GM12878 or HeLa CTCF HiChIP, suggesting our HiChIP experiments recalled PCa-specific CTCF loops.

9. The author claimed that the deletion of the proximal (-10kb) CRE perturbs MYC abundance and its relative chromatin affinity (Fig 3 c,d). They do not provide sufficient data supporting this conclusion: could they demonstrate by

WB that MYC protein abundance is increased? Could they provide data showing MYC chromatin binding is altered? How many binding sites do they find? What is the level and the distribution of MYC binding genome-wide? Which genomic region showed an increment of MYC binding? How did they determine MYC occupancy?

We agree with the reviewer on these points. To further assess the role of “-10Kb CTCF site” in the regulation of MYC expression and function, we performed additional validation experiments by targeting this CRE using dCas9-KRAB complex. We first confirmed CRISPRi of this CRE resulted in a dramatical reduction of the CTCF binding affinity at this site by ChIP-qPCR (**Figure S4f**). We then proved both mRNA and protein levels of MYC increased after repression of “-10Kb CTCF site” (**Figure S4g**). Four MYC target genes (CCDC86, CDC25A, NCL, and PCNA) were then selected for further validation based on their expression alteration in MYC overexpression/knockdown RNA-Seq data and MYC occupancy in MYC ChIP-Seq data. As determined by RT-qPCR analyses, mRNA levels of the four genes were all upregulated upon repression of “-10Kb CTCF site” (**Figure S4j**). Consistently, MYC binding affinity was also increased as shown in MYC ChIP-qPCR results (**Figure S4i**). These data support that repression of “-10Kb CTCF site” leads to the activation of MYC expression and increased binding affinity.

10. In fig. 3e,f the authors used TF binding motif analyses to conclude that “high MYC expression facilitate CTCF-CTCF loop formation”. They rely on the

different enrichment of MYC motif in the proximity of CTCF motifs within the CTCF-only anchors, with respect to CTCF and H3K27ac anchors. This kind of differential enrichment for a given motif can suffer a lot based on the composition and complexity of the compared genomic regions. Are the number of anchors comparable in these two samples? Is the level of CTCF binding comparable? If so how do they justify the result that indicate that CTCF motifs is differentially enriched in this comparison?

We agree with the reviewer that the motif analysis could be affected by many potential covariates. The motif co-localization can provide supporting evidence for the co-occupancy of MYC and CTCF, but has limited power to explain the binding affinity variation (The CTCF binding affinity regulation by MYC were further explored by experiments and analyses in **Figure 4**). Therefore, this section has been rephrased as: *“These results suggest MYC and CTCF motifs are co-localized at a subset of genomic regulatory elements, which is associated with the co-occupancy of MYC and CTCF at these sites.”*

11. The comparative analyses performed upon OE of MYC, brought to the general conclusion that MYC facilitate CTCF chromatin occupancy at the shared sites. Once again this is an over-interpretation of the obtained results. As shown in panel 4f, there is already a clear difference in the relative abundance of CTCF binding, depending which genomic regions have been considered, indicating that CTCF occupancy is different among the anchors, depending on the chromatin context (H3K27ac, in this case). The measured higher abundance of CTCF signal (from ChIP-seq) could depend by multiple regions, apart an increase in chromatin occupancy, which relates to an increase of the affinity and/or a change in the resident time. As all these results relies on population-based methodology, it may simply reflect that more cells at that given time of the analyses showed occupancy of CTCF. Plus, MYC alters multiple biological functions, including the cell cycle. Given that genome organization (including TADs) reflects replication timing, it is more that reasonable that MYC OE perturb the genome replication pattern (or time of division), thereby explaining the modes increment of CTCF binding.

We thank the reviewer for these thoughtful comments. Considering CTCF binding is much more increased at MYC-positive CTCF anchors compared with MYC-negative ones after MYC overexpression, and the co-localization of MYC and CTCF motifs at MYC-positive CTCF anchors, our data support MYC facilitates CTCF binding at shared sites. However, we agree with the reviewer that we could not rule out other mechanisms participating in the MYC-induced CTCF binding increase. We have updated it in our discussion section as follows:

“Although our data suggest MYC assists CTCF chromatin binding at MYC/CTCF common sites, we could not exclude additional function of MYC in

the regulation of CTCF chromatin occupancy. For example, considering CTCF chromatin binding was reported to display a cell cycle stage-dependent dynamics²⁶, and MYC is a well-established regulator in cell cycle control²⁷, MYC-regulated cell cycle progression may also play a role in CTCF binding changes.”

12. The Co-IP experiments (Fig 4h) lack important control to support the conclusion that MYC and CTCF are interacting. For example, do the author consider the possibility that chromatin bridging affected their results? The experimental conditions (buffer composition for example) are not clear, thereby is difficult to determine which ionic strength have been used in this assay. They need to include positive and negative controls to determine the specificity of this assay, including known MYC interactors (apart MAX). How do they explain that the OE of MYC did not change the level of immunoprecipitated MYC and CTCF? If they imply cooperativity between these two TFs, which domain/region of MYC and CTCF participate in this interaction? Finally orthogonal assays as to be included to support the conclusion that MYC and CTCF can interact within cells (like PLA or direct measurement of interaction like FRET experiments).

We appreciate the reviewer for the advice and now we have added the composition information for IP binding buffer to the Methods section. We also addressed the other points as below.

For the co-IP experiment in **Figure 4h**, the same amount of MYC antibody-conjugated beads were used for immunoprecipitation in control and MYC-OE samples. As the reviewer pointed out, the IPed MYC level was not significantly changed. The IPed MYC bands were all super strong, reflecting the saturation of co-IP signal. This usually happens when the assayed protein is the direct target of IP antibody. As a contrast, the level of co-IPed CTCF by MYC antibody-conjugated beads was higher in MYC-OE cells than control cells, which is consistent with MYC protein level change. Despite of the saturation in MYC bands, these results still support the MYC-CTCF interaction.

To further address the reviewer’s concerns of our Co-IP experiments, we performed multiple orthogonal assays to validate the protein-protein interaction of MYC and CTCF.

As shown above, we constructed GST-MYC full length fusion vector and expressed it in *E. coli*. The purified GST-MYC protein showed a clear interaction with MAX (MYC partner protein) and CTCF, while the negative control GST did not (**Figure 4i**).

We also constructed fusion vectors (GFP-MYC and Flag-CTCF) for mammalian cell expression and transfection. Co-IP experiments using the GFP antibody proved that exogenous MYC and CTCF proteins interact with each other in 22Rv1 cells (**Figure 4j**).

In the immunoprecipitation experiments, we have included IgG (**Figure 4h**), GST (**Figure 4i**) and GFP-EV (**Figure 4j**) as negative controls.

Finally, we performed proximity ligation assay (PLA) to assess the MYC-CTCF interaction within cells. The PLA result showed a clear MYC-CTCF protein interaction in the nuclei of 22Rv1 cells (**Figure 4k**).

13. Though correlative analyses, the author concluded that MYC suppresses neuroendocrine gene expression by enhancing CTCF chromatin looping (Fig 5). However they need to provide supportive data to this conclusion. For example, they should perturb CTCF (or MYC) binding to one of this anchor site by point mutation to then determine the relative change in gene expression. Is indeed expected from the claimed conclusion that the cooperation between MYC and CTCF on these specific genomic regions being causative of the increased chromatin looping and thereby the diminished gene expression. If so, the author should demonstrate this directly by perturbing the binding of these TFs on this site to then determine the relative change of chromatin looping and gene expression. This is the main point of their work, thereby they have to provide supportive (and not correlative) data.

We agree with this important suggestion from the reviewer. To assess the contribution of CTCF sites in MYC-repressed CDK5R2 expression, we individually targeted three CTCF sites at the anchors of new CTCF loops by dCas9-KRAB strategy. As shown above, CTCF ChIP-qPCR showed that CTCF binding affinity was significantly reduced at all three sites by CRISPRi (**Figure 5k**). Consequently, CDK5R2 mRNA levels went up in all three CRISPRi cell

lines under MYC overexpression condition. Since the strength of TF-associated looping is highly correlated with TF binding affinity (as illustrated in the **Reviewer figure 1**), these experimental data support our conclusion that MYC suppresses neuroendocrine gene transcription by enhancing CTCF-mediated chromatin looping.

Reviewer #3 (Remarks to the Author):

The main and most important finding in this study is that Myc, by directly interacting with CTCF, can alter the CTCF-mediated 3D genome organization in prostate cancer cell lines. The authors utilized multiple, publicly available genomics datasets, but also performed a great amount of HiChIP experiments and bioinformatic analyses.

Several pieces of information and a great number of conclusions that are presented here as novel have been previously published, either by the authors or other groups (MYC overexpression leads to increased chromatin interactions at superenhancers and MYC binding sites, doi:10.1101/gr.276313.121, Genome Res. 2022, or Transcriptional Dysregulation of MYC Reveals Common Enhancer-Docking Mechanism, <https://doi.org/10.1016/j.celrep.2018.03.056>). Therefore, it would be important to properly present what is already known in the field and nicely uncover the novel information of this manuscript.

We appreciate the reviewer for the great comments and thoughtful suggestions on how to improve our manuscript.

The following points would require further clarification in the manuscript:

1. Line 60: What CTCF interactome the authors refer to? Is it the protein interactome or the CTCF-dependent 3D genome organization of PCA cells?

We used CTCF interactome to refer to the CTCF-associated chromatin loops, which was enriched in CTCF HiChIP assay. To avoid misleading, we now changed “CTCF interactome” to “CTCF looping”.

2. For example a great number of findings, presented in the manuscript, have already been previously published:

- CTCF binding site located 2 kb upstream of the MYC promoter, <https://www.ncbi.nlm.nih.gov/entrez/eutils/elink.fcgi?dbfrom=pubmed&retmode=ref&cmd=prlinks&id=29641996>, <https://doi.org/10.1016/j.celrep.2018.03.056>
- Nucleic Acids Research, Volume 47, Issue 13, 26 July 2019, Pages 6699–6713, <https://doi.org/10.1093/nar/gkz462>

We agree with the reviewer that a number of studies have revealed the

regulation of MYC expression by chromatin looping or MYC's function in 3D genome organization and have cited a number of studies in our manuscript.

For the regulation of MYC expression by chromatin looping:

Ahmed, M. *et al.* CRISPRi screens reveal a DNA methylation-mediated 3D genome dependent causal mechanism in prostate cancer. *Nat Commun* **12**, 1781 (2021). **(Cited at line 72, line 293)**

Schuijers, J. *et al.* Transcriptional Dysregulation of MYC Reveals Common Enhancer-Docking Mechanism. *Cell Rep* **23**, 349-360 (2018). **(Cited at line 72, line 300)**

Cho, S.W. *et al.* Promoter of lncRNA Gene PVT1 Is a Tumor-Suppressor DNA Boundary Element. *Cell* **173**, 1398-1412 e22 (2018). **(Cited at line 72, line 449)**

Guo, H. *et al.* Androgen receptor and MYC equilibration centralizes on developmental super-enhancer. *Nat Commun* **12**, 7308 (2021). **(Cited at line 306, line 453)**

For MYC's function in 3D genome organization:

Kieffer-Kwon, K.R. *et al.* Myc Regulates Chromatin Decompaction and Nuclear Architecture during B Cell Activation. *Mol Cell* **67**, 566-578 e10 (2017). **(Cited at line 76, line 457)**

See, Y.X., Chen, K. & Fullwood, M.J. MYC overexpression leads to increased chromatin interactions at super-enhancers and MYC binding sites. *Genome Res* **32**, 629-642 (2022). **(Cited at line 78, line 459)**

To address the reviewer's comments, we emphasized in the revised manuscript that our novel finding is MYC promotes global CTCF-associated chromatin looping at line 428~429, as shown below.

"The novel finding in this study is MYC potentiates CTCF-mediated chromatin looping to suppress the expression of a subset of genes in PCa."

We also added the citation of a NAR paper to highlight the known mechanism about MYC expression regulation.

Hyle, J. *et al.* Acute depletion of CTCF directly affects MYC regulation through loss of enhancer-promoter looping. *Nucleic Acids Res* **47**, 6699-6713 (2019) **(Cited at line 451)**

3. More information would be required in the introduction regarding the use of DHT, or androgen induction. This would be helpful for following the results section.

DHT/androgen induction is a routine practice in prostate cancer cell lines, to examine androgen-elicited effects. We have added more details for the use of DHT in the Methods section, as below.

"For androgen stimulation treatment, cells were grown to 50%~60% confluence in a medium containing 5% charcoal-dextran stripped FBS (CDS) for 48 hr and then treated by 10nM DHT for 2 or 24 hr."

4. Fig.1h: the AR-ChIPseq, hours of DHT treatment is not indicated (2h or 24h?).

This AR ChIP-Seq data is from GSE55062. The treatment time is 12hr. We now added this information to the figure.

5. Fig.S1j,k: The fact that H3K27ac loops are changed or not would be better supported by quantitation instead of only visual examination. Quantitative data on loop-strength for these gene loci would be appreciated.

As advised, we have added the total normalized strength of H3K27ac loops to these panels.

6. Lines 166-170: please elaborate on the working hypothesis regarding the androgen-induced redistribution of cofactors. For serving the flow of the text in the manuscript, it is not clear why androgen-induced transcription repression is examined.

In our previous report, we documented that androgen-induced redistribution of cofactors underly the activation of AR-mediated sites and repression of H3K27ac-mediated enhancers (Guo et al., Nat Commun, 2021, PMID: 34911936). Here we proposed similar mechanisms are linked to AR-associated chromatin contacts (enriched at androgen-stimulated sites) versus H3K27ac-associated chromatin contacts (enriched at androgen-repressed sites). To help the transition and flow of our text, now we re-phrased the sentence as below:

“We previously reported redistribution of cofactors mediates androgen-activation of AR-binding sites and repression of H3K27ac-binding enhancers¹⁴. Similar mechanisms may engage with AR-associated chromatin contacts (enriched at androgen-stimulatory loci) versus the overall H3K27ac-associated chromatin contacts (enriched at androgen-repressive loci).”

7. Line 197: it is not described what the cell lines under study are, what is their origin, in order to understand the differences in HiChIP experiments. For example, why is there so much difference in the CTCF HiChIP results for VCaP and 22Rv1 cell lines?

We now added the origin of the two cell lines used in our CTCF HiChIP experiments at line 204~207, as below.

“To further delineate the function of CTCF in PCa, we generated CTCF-mediated chromatin contact maps by HiChIP in VCaP and 22Rv1, two widely used PCa cell lines derived from a vertebral metastasis of a prostate cancer patient and a human prostatic carcinoma xenograft, respectively.”

In addition, we have repeated our CTCF HiChIP experiments using another CTCF antibody (CST#3418S). Our new data confirmed that the difference in CTCF looping based on CTCF HiChIP datasets of two cell lines, by demonstrating a good reproducibility between the replicates for each cell line. Our new data were included in the updated manuscript.

8. Line 200: Figure 2a is called which is displaying ChIPseq results for CTCF and not HiChIP anchors. Please correct.

We apologize for the confusing. In previous **Figure 2a**, we compared both CTCF ChIP-Seq peaks and CTCF HiChIP loops between VCaP and 22Rv1 cells. The identity of a CTCF HiChIP loop is the combination of two anchors of this loop. Now we re-generated Venn diagram for CTCF HiChIP loops using our new CTCF HiChIP datasets. Because our new CTCF HiChIP analyses identified much more CTCF loops compared with previous one and we used the common CTCF loops of two replicates for overlapping analyses, there are more overlapping CTCF loops for VCaP and 22Rv1 cells in the new data compared with previous data. The ratio of overlapping is still lower for HiChIP loops compared with ChIP-Seq peaks, which is consistent with our previous **Figure 2a**. We now moved this panel to **Figure S3c** and modified the description as follows:

*“Only 44.92~63.63% of the CTCF HiChIP loops were common between VCaP and 22Rv1 cells (**Figure S3c**), while 75~78% of CTCF ChIP-Seq peaks were shared between VCaP and 22Rv1 cells (**Figure S3c**), suggesting CTCF HiChIP captures a higher cell-type heterogeneity compared with CTCF ChIP-Seq.”*
(Line 214~218)

9. Fig.2a: Please check graphs, they look almost the same although they depict CTCF loops in different cell lines.

As mentioned above, we now re-generated this panel and moved it to **Figure S3c**.

10. Fig.S2e: It is not clear what is depicted. Although, not H3K27ac ChIPseq was used but rather H3K27ac HiChIP experiment, is the overlap between K27ac anchors with CTCF anchors indicated? This is not clear in every case (Supplementary Fig. 1,2). Please clearly indicate what each figure depicts. Are they H3K27ac HiChIP anchors, or H3K27ac ChIPseq peaks? The same for all transcription factors used.

We apologize for not being clear. In previous Figure S2e, CTCF HiChIP loop anchors were used to count the anchor numbers, and these CTCF anchors were divided into H3K27ac+ and H3K27ac- subtypes by measuring the overlapping between CTCF anchors and H3K27ac ChIP-Seq peaks. We now

re-generated this figure in **Figure S3h** using our new CTCF HiChIP datasets and revised the figure legend accordingly. We also made further clarification for the datasets in the legends of updated **Figure S2g** and **S2h**.

11. Fig. S2f: The figure legend does not provide any information on how the promoter is defined so that distance categories are assigned. Based on this graph, over 80% of CTCF loop anchors are localized on genes and their promoters.

The anchor annotation was conducted by *annotatePeak* function from R package *ChIPseeker*. The promoter region was defined as 3Kb upstream and downstream of gene TSS, and further divided into ≤ 1 Kb, 1~2Kb and 2~3Kb regions as indicated. Other genomic region information was extracted from hg19 known genes of UCSC. We now used our new CTCF HiChIP datasets to repeat the analyses of previous Fig. S2f and got a similar result. We have added the information to the legend of new **Figure S3i**.

12. Lines 223-227, Fig. 2d: the conclusion drawn is an overestimation and the data presented do not support it. Please elaborate.

We agree with the reviewer for these comments. We now revised it as below:

*“Compared with H3K27ac-negative (H3K27ac -/-) CTCF loops, CTCF loops with double-positive H3K27ac in two anchors (H3K27ac +/+) were positively related with cell-type-specific gene expression in both 22Rv1 ($P = 1.3e-08$) and VCaP cells ($P < 2.2e-16$; **Figure 2f**), suggesting H3K27ac +/+ CTCF loops play a role in promoting gene transcription.”* (Line 238~242)

13. Fig.2e: the conclusion is not clear (lines 230-236). There is no cell-type specific enrichment of cancer-associated gene pathways for genes localized to CTCF HiChIP anchors (the criteria for proximity are not described).

We apologize for the inaccurate description. We now added the criteria for proximity to the legend of **Figure 2g**. Based on the KEGG results of our new data, the description has been updated as follows:

*“Many cancer-related pathways were enriched in H3K27ac +/- common CTCF loop anchors, such as “Pancreatic cancer” and “Acute myeloid leukemia” (**Figure 2g**). Interestingly, “Prostate cancer” pathway genes were enriched in both common and cell-type-specific CTCF loop anchors (**Figure 2g**), which is consistent with the tissue origin of VCaP and 22Rv1 cells, suggesting CTCF looping is involved in tissue-specific gene regulation.”* (Line 246~251)

14. Lines 286-287: The choice of the PCAT1/2 gene and the reference to its

super-enhancers is not justified.

We apologize for the insufficient explanation. In our previous report (Guo et al., Nat Commun, 2021, PMID: 34911936), we identified three super-enhancers within PCAT1/2 region based on H3K27ac ChIP-seq peaks by Ranking Of Super Enhancer (ROSE) and performed 3C-ddPCR to validate these super-enhancers interact with MYC promoter to regulating MYC expression in VCaP cells. Our findings are consistent with reports that in prostate cancer cells (LNCaP and VCaP) the PCAT1/2 sites physically and functionally interact with MYC promoter (PMID: 20453196; PMID: 31735626). In current study, this distal chromatin interaction was reproduced in 22Rv1 cells. Now, we added a citation of our previous work to introduce these super-enhancers at line 304~308 (as shown below).

“We and others previously reported that a cluster of super-enhancers within PCAT1/2 region interact with MYC promoter to regulate MYC expression in VCaP cells^{14,21,22}. Here, we found the super-enhancers within PCAT1/2 region were also robustly looped to MYC in 22Rv1 cells, but the looping between them was not significantly changed by -10Kb CTCF site deletion (Figure S4d).”

15. Fig. S3c: Myc locus is localized on chromosome 8, though upon deletion of the -10Kb element, bigger effects are observed in CTCF looping in other chromosomes compared to the anticipated cis- effects. Taken under consideration the Myc hypothesis, one would expect in Fig.3c to present an analysis of the CTCF loops presented in SFig.3c. Moreover, in Fig.3d is there really an obvious difference in Myc binding in CTCF anchors in the samples +/- the -10Kb deletion?

We agree with the reviewer for the figure arrangement. **Figure 3c** has now been exchanged with previous **Figure S3c**.

We also agree with the reviewer that the difference in **Figure 3d** is not dramatic. This section has been toned down as rephrased as:

“Since the primary effect of -10Kb CTCF site deletion is to drive MYC upregulation, we speculated the global alteration of CTCF loops is associated with chromatin occupation of MYC. In agreement with our hypothesis, MYC binding sites were enriched in both sgDele-10Kb-specific and sgCtrl-specific CTCF anchors (Figure S4e and 3d).” (Line 324~328)

16. HiChIP protocol: Restriction enzyme digestion takes place for only 20 minutes?

We apologize for this error. The digestion time should be 2 hours. We have

corrected it in the Methods section.

Immunoprecipitation takes place in 1% SDS (Nuclear Lysis Buffer)?

The lysate in Nuclear Lysis Buffer would be further diluted before immunoprecipitation that takes place in 0.1% SDS. We have added the following sentence in the Method section:

“After sonication, the lysate was diluted to 1:9 by adding ChIP dilution buffer (0.01% SDS, 1.10% Triton X-100, 1.2 mM EDTA, 16.7 mM Tris-HCl pH 7.5, 167 mM NaCl).”

Reviewers' Comments:

Reviewer #1:

Remarks to the Author:

The authors have successfully addressed my concerns. The manuscript is ready for publication.

Reviewer #2:

None

Reviewer #3:

Remarks to the Author:

Taking under consideration the comments of all 3 reviewers and the great amount of effort the authors have put to address any outstanding issues, I believe that most comments have been appropriately answered. Therefore, my suggestion is to accept the manuscript for publication.

REVIEWERS' COMMENTS

Reviewer #1 (Remarks to the Author):

The authors have successfully addressed my concerns. The manuscript is ready for publication.

We thank the reviewer for the in-depth-comments and suggestions, which improved the manuscript.

Reviewer #3 (Remarks to the Author):

Taking under consideration the comments of all 3 reviewers and the great amount of effort the authors have put to address any outstanding issues, I believe that most comments have been appropriately answered. Therefore, my suggestion is to accept the manuscript for publication.

We thank the reviewer for the in-depth-comments and corrections, which improved the manuscript.